# LIGHTWEIGHT IMAGE SUPER-RESOLUTION VIA FLEXIBLE META PRUNING

## ABSTRACT

Lightweight image super-resolution (SR) methods have obtained promising results with moderate model complexity. These approaches primarily focus on a lightweight architecture design, but neglect to further reduce network redundancy. While some model compression techniques try to achieve more lightweight SR models with neural architecture search, knowledge distillation, or channel pruning, they typically require considerable extra computational resources or neglect to prune weights. To address these issues, we propose a flexible meta pruning (FMP) for lightweight image SR, where the network channels and weights are pruned simultaneously. Specifically, we control the network sparsity via channel vectors and weight indicators. We feed them into a hypernetwork, whose parameters act as meta-data for the parameters of the SR backbone. Consequently, for each network layer, we conduct structured pruning with channel vectors, which control the output and input channels. Besides, we conduct unstructured pruning with weight indicators to influence the sparsity of kernel weights, resulting in flexible pruning. During pruning, the sparsity of both channel vectors and weight indicators are regularized. We optimize the channel vectors and weight indicators with proximal gradient and SGD. We conduct extensive experiments to investigate critical factors in the flexible channel and weight pruning for image SR, demonstrating the superiority of our FMP when applied to baseline image SR architectures. Code and pretrained models will be released.

## 1 INTRODUCTION

As one of the fundamental image processing tasks, single image super-resolution (SR) aims to upscale a low-resolution (LR) input to the desired size by restoring more details. Recently, the task has received increasing attention, with much exploration of deep neural network architectures for improved performance and efficiency (Dong et al., 2014; Kim et al., 2016a; Lim et al., 2017; Zhang et al., 2018b; Liang et al., 2021). Image SR first witnessed the application of deep convolutional neural networks (CNN) in SRCNN (Dong et al., 2014), with three convolutional layers. Kim *et al.* successfully trained a deeper network with residual learning (Kim et al., 2016a). Lim *et al.* further built a much deeper network EDSR (Lim et al., 2017) by simplifying the residual blocks (He et al., 2016). Zhang *et al.* achieved even deeper in RCAN (Zhang et al., 2018a) with the residual in residual (RIR) structure. Such an RIR structure was further utilized in SwinIR (Liang et al., 2021), where Swin Transformer (Liu et al., 2021) was introduced as the basic block. Most of these CNN and Transformer based methods have obtained increasing SR performance with large model size and run-time, making them hard to deploy in practice. Therefore, lightweight models are heavily desired in real-world applications, where the computational resources are limited (Lee et al., 2020).

To achieve lightweight image SR models, increasing effort has been devoted to design lightweight architectures and incorporation of model compression techniques. Many well-designed lightweight SR models have been proposed, such as CARN (Ahn et al., 2018), IMDN (Hui et al., 2019), RLFN (Kong et al., 2022), and ELAN (Zhang et al., 2022a). However, the required architectural exploration is costly in both time and energy. Meanwhile, knowledge distillation (KD) (Hinton et al., 2014) was introduced to distill knowledge from a teacher SR network to the student (He et al., 2020; Lee et al., 2020). Meanwhile, neural architecture search (NAS) (Zoph & Le, 2017; Elsken et al., 2019) was also utilized to explore lightweight SR structures, including MoreMNAS (Chu et al., 2019b) and FALSR (Chu et al., 2019a). However, they also have several drawbacks. KD-based

methods usually require a large teacher network, consuming considerable computational resources. NAS-based methods often need a large computational budget for searching. Most manually designed, distilled, or searched lightweight networks further neglect to deeply consider inference time by also removing redundant network channels and weights. This could be suitably conducted by network pruning techniques (*i.e.*, structured and unstructured pruning).

To explore the network redundancy and reduce complexity, researchers usually turn to network pruning techniques (Reed, 1993; Sze et al., 2017), mainly consisting of structured pruning (*i.e.*, channel pruning) (Li et al., 2017) and unstructured pruning (*i.e.*, weight pruning) (Han et al., 2015; 2016). Aligned structured sparsity was investigated and jointly optimized in image SR network ASSLN (Zhang et al., 2021; 2022b). They paid much attention to aligning pruned channel locations across different layers. On the other hand, MetaPruning (Liu et al., 2019) learned the parameters of the backbone network via hypernetworks (Ha et al., 2017), which can only obtain fixed-size weights. However, such outputs can hardly be used for layer-wise configuration searching. To control the output size, Li *et al.* proposed a differentiable meta pruning method via hypernetworks (DHP) (Li et al., 2020a). However, DHP only prunes the network channels for image SR and neglects to remove redundant kernel weights.

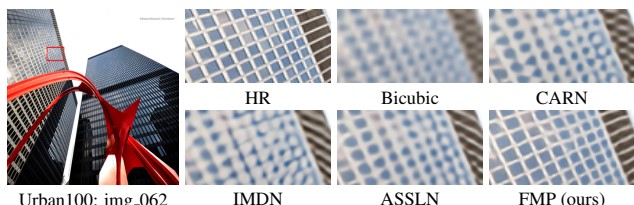

Figure 1: Visual samples of image SR ($\times 4$) by lightweight methods. Our FMP achieves better visual reconstruction.

To address the aforementioned problems, we first design a lightweight SR baseline (LSRB) and then propose a flexible meta pruning (FMP) technique. Specifically, following the NTIRE 2022 Challenge on Efficient Super-Resolution (ESR) (Li et al., 2022), we primarily focus on actual inference time. We design LSRB based on residual blocks (Lim et al., 2017) and enhanced spatial attention (ESA) (Liu et al., 2020). We then propose a hypernetwork, whose parameters serve as meta-data for those of the backbone network. The hypernetwork takes channel vectors and weight indicators as inputs and obtains network parameters for the SR backbone LSRB without pretraining. The channel vectors control the output and input channels of each network layer for structured pruning. While, weight indicators influence the sparsity of kernel weights for unstructured pruning. We adopt a sparsity regularizer to channel vectors and weight indicators, resulting in automatic network pruning. In the pruning stage, we optimize the channel vectors and weight indicators with proximal gradient and SGD, respectively. This stage is then halted when the target compression ratio is reached. After the pruning stage, the channel vectors and weight indicators are sparsified. The corresponding channels and kernel weights of the backbone network are also pruned flexibly.

The main contributions are summarized as follows:

- We propose a flexible meta pruning (FMP) technique for lightweight image super-resolution (SR). We jointly conduct flexible structured and unstructured network pruning during the image SR training.
- We propose channel vectors and weight indicators to control backbone channel and weight sparsity. We optimize them with proximal gradient and SGD, respectively, enabling differentiable and flexible pruning.
- We design a simple yet effective SR baseline (LSRB), achieving better performance than the champion solution in ESR challenge. We apply our FMP to LSRB and other baselines, obtaining more efficient backbones and showing the effectiveness of FMP (see Fig. 1).

## 2 RELATED WORKS

**Lightweight Image SR Networks**. Recently, lightweight image SR networks have been attracting consistent attention and achieved promising performance. Dong *et al.* proposed FSRCNN (Dong et al., 2016) to accelerate image SR by placing the upscale module at the tail network position. Ahn *et al.* proposed CARN (Ahn et al., 2018) with a cascading mechanism in a residual network. Hui *et al.* constructed the cascaded information multi-distillation for a lightweight network (IMDN) (Hui et al., 2019). Kong *et al.* proposed a residual local feature network (RLFN) (Kong et al., 2022) with enhanced spatial attention (ESA) (Liu et al., 2020). Zhang *et al.* proposed an efficient long-range attention network (ELAN) (Zhang et al., 2022a). Also, model compression techniques have been utilized for lightweight image SR. He *et al.* (He et al., 2020) and Lee *et*

*al.* introduced knowledge distillation (Hinton et al., 2014) and proposed to learn with privileged information (Lee et al., 2020). In the meantime, Chu *et al.*introduced neural architecture search (NAS) (Zoph & Le, 2017; Elsken et al., 2019) for image SR in FALSR (Chu et al., 2019a) and MoreMNAS (Chu et al., 2019b). Although those works have achieved notable progress, they still need to carefully design the architectures or consume extra resources.

**Network Pruning**. In the deep networks, there is considerable number of redundant parameters, which could be pruned without hurting performance too much (Reed, 1993; Sze et al., 2017; Cheng et al., 2018a;b). Network pruning techniques could be roughly divided into structured pruning (*i.e.*, channel pruning) (Li et al., 2017; Wen et al., 2016; He et al., 2017; Wang et al., 2021) and unstructured pruning (*i.e.*, weight pruning) (Han et al., 2015; 2016). With a pretrained large model, Zhang *et al.* integrated channel pruning into image SR with aligned structured sparsity in ASSL (Zhang et al., 2021) or structure-regularized pruning in SRP (Zhang et al., 2022b), which utilized pretrained models. Structured pruning usually leads to regular sparsity after pruning. While, unstructured pruning produces irregular sparsity (Wen et al., 2016; Wang et al., 2019). Very few image SR works investigate such things. In this work, we focus on a flexible pruning technique, which considers both channel and weight pruning simultaneously without any pretrained networks via a hypernetwork.

**Meta Learning**. As a concept of learning to learn, meta learning is a wide collection of machine learning methods. It was also introduced in image SR (Hu et al., 2019), where the meta-upscale module was proposed to dynamically predict the weights of the upscale filters for the arbitrary scaling factor. Recently, one hot meta learning topic has been about using a hypernetwork (Ha et al., 2017) to predict the weight parameters of the backbone network. Such ideas about hypernetwork have been widely investigated in NAS (Brock et al., 2018), network channel pruning (Liu et al., 2019; Li et al., 2020b), and image super-resolution (*e.g.*, DHP (Li et al., 2020a)). However, most of them focus on channel pruning and neglect to prune the weights. In this work, we design a more general hypernetwork, which deals with channel pruning and weight pruning for each network layer simultaneously and obtains more efficient image SR networks.

## 3 PROPOSED METHOD

### 3.1 MOTIVATION

**Why Flexible Pruning?** The general idea of this work is first introduced before elaborating on details of our flexible network pruning method. Structured pruning and unstructured pruning are two important network compression methods that can cut down the model complexity of deep neural networks significantly. They have different strengths. On one hand, structured pruning leaves regular kernels after pruning, which is beneficial for the actual acceleration of the network. On the other hand, unstructured pruning removes single weights in a kernel and can compress the network without sacrificing too much accuracy of the network. However, the unstructured pruning leads to irregular kernels, which can hardly reduce time. They need specific hardware designs to achieve actual acceleration. Thus, coupling structured pruning and unstructured pruning brings together the merits of both techniques, squeeze out the redundancy from deep network, and take full advantage of the capacity of the network under a fixed budget.

**How to do Flexible Pruning?** We design a flexible network pruning method that shrinks the model by the specified compression ratio, using structured and unstructured pruning. An important problem that follows is how to couple the two techniques during the design of the algorithm, while at the same time decouple them during the optimization of the pruning process. To achieve that, we propose to utilize hypernetworks that, in short, predict the parameters of the backbone network. The key components of the proposed method are the channel vectors and the weight indicator, which serve as tools to handle structured pruning and unstructured pruning. Each convolutional layer of the backbone network is assigned a channel vector and a weight indicator. The channel vectors control the number of output channels of the convolutional layer, while the weight indicators reflect the influence of single weight parameters of the network. By manipulating the channel vectors and the weight indicator during the optimization of the pruning process, we achieve joint structured and unstructured pruning (see Fig. 2).

### 3.2 LIGHTWEIGHT SR BASELINE

Deep image SR networks learn a mapping from a low-resolution (LR) image $I_{LR}$ to its high-resolution (HR) counterpart $I_{HR}$. Here, we focus on lightweight SR networks, which have fewer parameters and computation operations, but achieve comparable or higher performance.

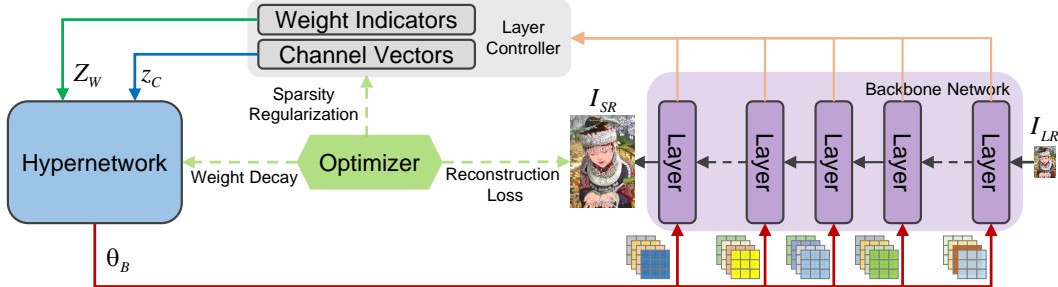

Figure 2: Illustration of our flexible meta pruning (FMP) for image SR. Each backbone layer is associated to layer controller, which provides channel vectors and weight indicators to hypernetwork. Its output $\Theta_B$ can actually serve as the weights of SR backbone. We optimize the whole pipeline by utilizing reconstruction loss, weight decay, and sparsity regularization.

Most of the previous lightweight image SR networks (Ahn et al., 2018; Hui et al., 2019) focus on reducing parameters and FLOPs. Although they have achieved high performance, their inference speed is usually not very fast, hindering their practical usage. Pursuing faster inference speed is attracting increasing attention. Recently, NTIRE 2022 Efficient Super-Resolution (ESR) workshop (Li et al., 2022) targets to investigate efficient SR models in terms of inference time and lightweight networks. Its latest champion solution is RLFN (Kong et al., 2022), which utilizes residual blocks (Lim et al., 2017) and enhanced spatial attention (ESA) (Liu et al., 2020) as building blocks (*i.e.*, RLFB).

In RLFN (Kong et al., 2022), its residual block consists of 3 convolutional (Conv) layers, each of which is followed by ReLU (Nair & Hinton, 2010). This can introduce too much non-linearities, which may hamper pixel-wise reconstruction tasks, such as image SR. On the other hand, we find that ESA performs better in lightweight networks than in large ones. It motivates us to use more ESA and fewer ReLU modules with

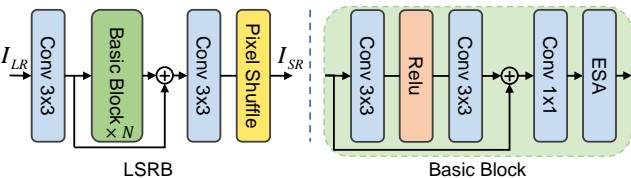

Figure 3: Framework of our designed LSRB based on RLFN (Kong et al., 2022). Left: There are $N$ basic blocks in LSRB. Right: Each basic block consists of residual block (Lim et al., 2017) (two Conv layers and one ReLU), Conv $1\times1$, and ESA (Liu et al., 2020).

limited model size. Consequently, we modify RLFB (Kong et al., 2022) by replacing its residual block with the simplified residual block (Lim et al., 2017), which has 2 Conv layers and a ReLU between them. We name this version as an lightweight SR baseline (LSRB) and show its details in Fig. 3. As investigated in our experiments, this simple LSRB can further reduce inference time with comparable performance. LSRB can also be used as backbone network and be further pruned by our proposed flexible meta pruning (see Fig. 2).

### 3.3 GENERAL HYPERNETWORK

**Notation**. Before delving deep into the details of the hypernetwork design, we first introduce the notation that is used throughout this paper. Let $c_{out}\times c_{in}\times k_1\times k_2$ denote the original kernel size and $c_{out}^p\times c_{in}^p\times k_1\times k_2$ denote the target kernel size in general. The corresponding original and target kernel sizes in $l$-th layer of the backbone network are denoted as $c_{out}^l\times c_{in}^l\times k_1\times k_2$ and $c_{out}^{p,l}\times c_{in}^{p,l}\times k_1\times k_2$. By compressing the weight parameters $W_B^l$ in the original network, we aim at to derive a compact representation of those weights, *i.e.*, $W_B^{P,l}$. To control the pruning of the network, we introduce another two tools, namely, the channel vectors $z_C^l \in \mathbb{R}^{c_{out}^l}$ and the weight indicators $Z_W^l \in \mathbb{R}^{c_{out}^l \times c_{in}^l \times k_1^l \times k_2^l}$. The size of the channel vector is equal to the number of output channels of the backbone layer and controls the pruning of the single channels. The weight indicator is initialized as a tensor with all ones and acts as a continuous mask that reflects the single weight strength. The design of the hypernetwork is inspired by (Li et al., 2020a) and tailored to the joint optimization of structured pruning and unstructured pruning in this work.

**Hypernetwork**. Following the design in DHP, the hypernetwork has three inputs including the channel vector of the previous layer, the channel vectors of the current layer, and the weight indicators of the current layer. The computation in the hypernetwork is conducted in three steps.

Step 1 │ A matrix, *i.e.*, $M^l = z_C^l \cdot (z_C^{l-1})^T$, forms a grid used for structured purning.

Step 2 │ Every element of the computed matrix is transformed to a vector by two linear operations

$$O_{i,j}^l = W_2^l \cdot (M_{i,j}^l \cdot W_1^l),\tag{1}$$

where $W_1^l \in \mathbb{R}^{m \times 1}$ and $W_2^l \in \mathbb{R}^{k^2 \times m}$, the scalar $M_{i,j}^l$ is the $i,j$-th element of the matrix $M^l$, and $O_{i,j}^l \in \mathbb{R}^{k^2}$, $O^l \in \mathbb{R}^{c_{out}^l \times c_{in}^l \times k^2}$. Note that for each element $M_{i,j}^l$, $W_1^l$ and $W_2^l$ are different and for the simplicity of notation the subscript $i,j$ is omitted.

Step 3 │ The output $O^l$ from the second stage is reshaped into $Z_C^l \in \mathbb{R}^{c_{out}^l \times c_{in}^l \times k \times k}$ and masked by the weight indicators. This results in the modified tensor

$$Z^l = Z_C^l \odot Z_W^l,\tag{2}$$

where $\odot$ denotes element-wise multiplication, $Z^l$ is the final output of the hypernetwork. The tensor $Z^l$ could be used as the weight parameters of the convolutional (Conv) layers of the SR backbone.

As a summary, we denote the parameters of the hypernetwork as $\Theta_H = \{W_1^l, W_2^l\}$. And the functionality of the hypernetwork can be simplified as

$$\Theta_B = \mathcal{F}_H(z_C, Z_W; \Theta_H),\tag{3}$$

where $\mathcal{F}_H(\cdot)$ denotes the hypernetwork function, $\Theta_B$ denotes the weights of the backbone network.

By optimizing channel vectors and weight indicators, it is possible to achieve structured pruning and unstructured pruning simultaneously (Fig. 4). Specifically, the network weights could be flexibly pruned along the channel (*i.e.*, output and in channels) and spatial dimensions. By the above formulation, we have already coupled structured pruning and unstructured pruning within the same framework. In next section, we deal with the problem of decoupling their optimization.

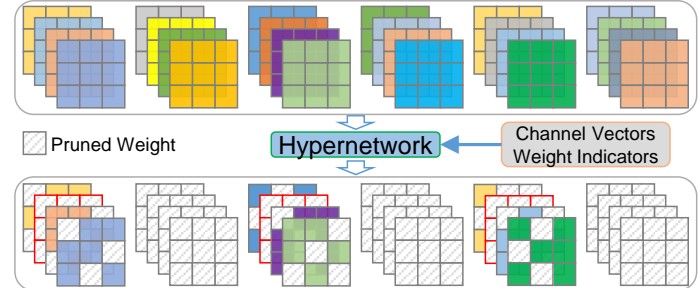

Figure 4: Illustration of hypernetwork in flexible meta pruning. Channel vectors and weight indicators are inputs.

### 3.4 FLEXIBLE META PRUNING FOR IMAGE SR

We show how to apply FMP for image SR in Fig. 2 and give more details about layer controller sparsity regularization and joint optimization during pruning in SR.

**Sparsity Regularization**. For the joint optimization of the network pruning and image SR problem, we use the following loss function, that contains four terms

$$\mathcal{L} = \mathcal{L}_{rec}(I_{HR}, I_{SR}) + \alpha \mathcal{D}(\Theta_H) + \lambda_C \mathcal{R}(z_C) + \lambda_W \mathcal{R}(Z_W),\tag{4}$$

where $\alpha$, $\lambda_C$, and $\lambda_W$ are regularization factors.

Our pruning algorithms is jointly optimized with image SR by considering the image reconstruction loss $\mathcal{L}_{rec}$ in the loss function. We use $L_1$ loss between the ground-truth HR image $I_{HR}$ and the reconstructed SR image $I_{SR}$. And the super-resolved output could be obtained via

$$I_{SR} = \mathcal{F}_{FMP}(I_{LR}; \mathcal{F}_H(z_C, Z_W; \Theta_H)).\tag{5}$$

The term $\mathcal{D}(\Theta_H)$ in Eq. 4 denotes the weight decay regularization applied to the parameters of the hypernetwork and $\alpha$ is the weight decay factor.

The network pruning is achieved by applying sparsity regularization on the channel vectors and the weight indicators which correspond to the terms $\mathcal{R}(z_C)$ and $\mathcal{R}(Z_W)$

$$\mathcal{R}(z_C) = \sum_{l=1}^{L} \left\| z_C^l \right\|_1,\tag{6}$$

$$\mathcal{R}(Z_W) = \sum_{l=1}^{L} \left\| Z_W^l \right\|_p.\tag{7}$$

In Eq. 7, the $L_p$ norm is applied as a regularization on the weight indicator. In the experiments, we ablate different choices of the norm (*e.g.*, $L_1$, $L_2$, and weight decay).

**Differentiable Optimization**. Since the channel vectors and the weight indicator are not intertwined by non-linear operations, we can decouple their optimization for network pruning by different methods. First, the parameters $\Theta_H$ in the hypernetwork is optimized by SGD,

$$\Theta_H[t+1] = \Theta_H[t] - \eta\nabla\mathcal{H}(\Theta_H[t]), \tag{8}$$

where $\mathcal{H} = \mathcal{L}_{rec} + \alpha\mathcal{D}$, and $\eta$ denotes the learning rate.

Second, for the optimization of the channel vectors, we apply proximal gradient descent method which contains a gradient descent step and a proximal step.

$$z_C[t+\Delta] = z_C[t] - \eta\nabla\mathcal{G}(z_C[t]), \tag{9}$$

$$z_C[t+1] = \mathbf{prox}_{\lambda\mu\mathcal{R}}\Big(z_C[t+\Delta] - \lambda\mu\nabla\mathcal{L}\big(z_C[t+\Delta]\big)\Big), \tag{10}$$

where $\mathcal{G} = \mathcal{L}_{rec} + \lambda_C\mathcal{R}$. $\mu$ is the step size of proximal gradient and is set as the learning rate of SGD update. The proximal operator of $L_1$ sparsity regularization on the channel vectors have closed-form solutions as the soft-thresholding function. Finally, the weight indicators are also optimized by the standard SGD, *i.e.*, $Z_W[t+1] = Z_W[t] - \eta\nabla\mathcal{E}(Z_W[t])$, where $\mathcal{E} = \mathcal{L}_{rec} + \lambda_W\mathcal{R}$.

During the optimization, we set compression ratio targets $\gamma_C$=0.1 and $\gamma_W$=0.02 for both structured pruning and unstructured pruning. If the compression ratio of one pruning method is achieved, the optimization method for either the channel vectors or the weight indicators is stopped while the other one continues. The whole algorithm converges if both of the compression targets are achieved.

## 4 EXPERIMENTAL RESULTS

### 4.1 SETTINGS

**Data and Evaluation.** Following most recent works (Timofte et al., 2017; Lim et al., 2017; Haris et al., 2018), we use DIV2K dataset (Timofte et al., 2017) and Flickr2K (Lim et al., 2017) as training data. We use five standard benchmark datasets: Set5 (Bevilacqua et al., 2012), Set14 (Zeyde et al., 2010), B100 (Martin et al., 2001), Urban100 (Huang et al., 2015), and Manga109 (Matsui et al., 2017). We evaluate the SR results with PSNR and SSIM (Wang et al., 2004) on Y channel of transformed YCbCr space. It should be noted that to obtain our results we do not use *self-ensemble*. We also provide model size and FLOPs comparisons. If not specifically stated, in the main comparison, we set the output size as $3\times1280\times720$ to calculate FLOPs.

**Training Settings.** Following (Lim et al., 2017; Zhang et al., 2018a), we perform data augmentation on the training images, which are randomly rotated by $90°$, $180°$, $270°$ and flipped horizontally. Each training batch consists of 16 LR color patches, whose size is $64\times64$. Our FMP model is trained by ADAM optimizer (Kingma & Ba, 2015) with $\beta_1$=0.9, $\beta_2$=0.999, and $\epsilon$=$10^{-8}$. We set the initial learning rate as $10^{-4}$ and then decrease it to half every $2\times10^5$ iterations. We use PyTorch (Paszke et al., 2017) to implement our models with RTX 3090 GPUs.

### 4.2 MAIN COMPARISONS

We apply FMP to LSRB and compare with representative lightweight SR networks: SRCNN (Dong et al., 2014), FSRCNN (Dong et al., 2016), VDSR (Kim et al., 2016a), DRCN (Kim et al., 2016b), LapSRN (Lai et al., 2017), DRRN (Tai et al., 2017a), MemNet (Tai et al., 2017b), CARN (Ahn et al., 2018), IMDN (Hui et al., 2019), LatticeNet (Luo et al., 2022), and ASSLN (Zhang et al., 2021). We configure LSRB to keep similar model size and FLOPs as recent leading ones (*e.g.*, IMDN).

**Quantitative Results**. In Tab. 1, we provide our quantitative results *without self-ensemble*. ASSLN (Zhang et al., 2021) ranks the second best place, while our FMP performs the best on all datasets across all scales. Specifically, let us take the high-quality Urban100 as an example. Our FMP obtains about 0.0051, 0.0035, and 0.0047 SSIM gains on Urban100 ($\times2$, $\times3$, $\times4$) over the second-best method, respectively. These comparisons show the effectiveness of FMP, which conducts flexible network pruning and increases the efficiency of the network parameters from hypernetwork. We make better use of the channel and weight sparsity of the backbone and increase efficiency of the network parameters from hypernetwork.

| Method | Scale | Set5 PSNR | Set5 SSIM | Set14 PSNR | Set14 SSIM | B100 PSNR | B100 SSIM | Urban100 PSNR | Urban100 SSIM | Manga109 PSNR | Manga109 SSIM |
|---|---|---|---|---|---|---|---|---|---|---|---|
| SRCNN (Dong et al., 2014) | ×2 | 36.66 | 0.9542 | 32.42 | 0.9063 | 31.36 | 0.8879 | 29.50 | 0.8946 | 35.60 | 0.9663 |
| FSRCNN (Dong et al., 2016) | ×2 | 37.00 | 0.9558 | 32.63 | 0.9088 | 31.53 | 0.8920 | 29.88 | 0.9020 | 36.67 | 0.9710 |
| VDSR (Kim et al., 2016a) | ×2 | 37.53 | 0.9587 | 33.03 | 0.9124 | 31.90 | 0.8960 | 30.76 | 0.9140 | 37.22 | 0.9750 |
| DRCN (Kim et al., 2016b) | ×2 | 37.63 | 0.9588 | 33.04 | 0.9118 | 31.85 | 0.8942 | 30.75 | 0.9133 | 37.63 | 0.9740 |
| LapSRN (Lai et al., 2017) | ×2 | 37.52 | 0.9590 | 33.08 | 0.9130 | 31.80 | 0.8950 | 30.41 | 0.9100 | 37.27 | 0.9740 |
| DRRN (Tai et al., 2017a) | ×2 | 37.74 | 0.9591 | 33.23 | 0.9136 | 32.05 | 0.8973 | 31.23 | 0.9188 | 37.92 | 0.9760 |
| MemNet (Tai et al., 2017b) | ×2 | 37.78 | 0.9597 | 33.28 | 0.9142 | 32.08 | 0.8978 | 31.31 | 0.9195 | 37.72 | 0.9740 |
| CARN (Ahn et al., 2018) | ×2 | 37.76 | 0.9590 | 33.52 | 0.9166 | 32.09 | 0.8978 | 31.92 | 0.9256 | 38.36 | 0.9764 |
| IMDN (Hui et al., 2019) | ×2 | 38.00 | 0.9605 | 33.63 | 0.9177 | 32.19 | 0.8996 | 32.17 | 0.9283 | 38.87 | 0.9773 |
| LatticeNet (Luo et al., 2022) | ×2 | 38.06 | 0.9607 | 33.70 | 0.9187 | 32.20 | 0.8999 | 32.25 | 0.9288 | N/A | N/A |
| ASSLN (Zhang et al., 2021) | ×2 | 38.12 | 0.9608 | 33.77 | 0.9194 | 32.27 | 0.9007 | 32.41 | 0.9309 | 39.12 | 0.9781 |
| FMP (ours) | ×2 | 38.17 | 0.9615 | 33.81 | 0.9215 | 32.32 | 0.9022 | 32.71 | 0.9360 | 39.17 | 0.9783 |
| SRCNN(Dong et al., 2014) | ×3 | 32.75 | 0.9090 | 29.28 | 0.8209 | 28.41 | 0.7863 | 26.24 | 0.7989 | 30.48 | 0.9117 |
| FSRCNN (Dong et al., 2016) | ×3 | 33.16 | 0.9140 | 29.43 | 0.8242 | 28.53 | 0.7910 | 26.43 | 0.8080 | 31.10 | 0.9210 |
| VDSR (Kim et al., 2016a) | ×3 | 33.66 | 0.9213 | 29.77 | 0.8314 | 28.82 | 0.7976 | 27.14 | 0.8279 | 32.01 | 0.9340 |
| DRCN (Kim et al., 2016b) | ×3 | 33.82 | 0.9226 | 29.76 | 0.8311 | 28.80 | 0.7963 | 27.15 | 0.8276 | 32.31 | 0.9360 |
| DRRN (Tai et al., 2017a) | ×3 | 34.03 | 0.9244 | 29.96 | 0.8349 | 28.95 | 0.8004 | 27.53 | 0.8378 | 32.74 | 0.9390 |
| MemNet (Tai et al., 2017b) | ×3 | 34.09 | 0.9248 | 30.00 | 0.8350 | 28.96 | 0.8001 | 27.56 | 0.8376 | 32.51 | 0.9369 |
| CARN (Ahn et al., 2018) | ×3 | 34.29 | 0.9255 | 30.29 | 0.8407 | 29.06 | 0.8034 | 28.06 | 0.8493 | 33.50 | 0.9539 |
| IMDN (Hui et al., 2019) | ×3 | 34.36 | 0.9270 | 30.32 | 0.8417 | 29.09 | 0.8046 | 28.17 | 0.8519 | 33.61 | 0.9444 |
| LatticeNet (Luo et al., 2022) | ×3 | 34.40 | 0.9272 | 30.32 | 0.8416 | 29.10 | 0.8049 | 28.19 | 0.8513 | N/A | N/A |
| ASSLN (Zhang et al., 2021) | ×3 | 34.51 | 0.9280 | 30.45 | 0.8439 | 29.19 | 0.8069 | 28.35 | 0.8562 | 34.00 | 0.9468 |
| FMP (ours) | ×3 | 34.55 | 0.9291 | 30.48 | 0.8456 | 29.20 | 0.8101 | 28.40 | 0.8597 | 34.06 | 0.9473 |
| SRCNN(Dong et al., 2014) | ×4 | 30.48 | 0.8628 | 27.49 | 0.7503 | 26.90 | 0.7101 | 24.52 | 0.7221 | 27.58 | 0.8555 |
| FSRCNN (Dong et al., 2016) | ×4 | 30.71 | 0.8657 | 27.59 | 0.7535 | 26.98 | 0.7150 | 24.62 | 0.7280 | 27.90 | 0.8610 |
| VDSR (Kim et al., 2016a) | ×4 | 31.35 | 0.8838 | 28.01 | 0.7674 | 27.29 | 0.7251 | 25.18 | 0.7524 | 28.83 | 0.8870 |
| DRCN (Kim et al., 2016b) | ×4 | 31.53 | 0.8854 | 28.02 | 0.7670 | 27.23 | 0.7233 | 25.14 | 0.7510 | 28.98 | 0.8870 |
| LapSRN (Lai et al., 2017) | ×4 | 31.54 | 0.8850 | 28.19 | 0.7720 | 27.32 | 0.7280 | 25.21 | 0.7560 | 29.09 | 0.8900 |
| DRRN (Tai et al., 2017a) | ×4 | 31.68 | 0.8888 | 28.21 | 0.7720 | 27.38 | 0.7284 | 25.44 | 0.7638 | 29.46 | 0.8960 |
| MemNet (Tai et al., 2017b) | ×4 | 31.74 | 0.8893 | 28.26 | 0.7723 | 27.40 | 0.7281 | 25.50 | 0.7630 | 29.42 | 0.8942 |
| CARN (Ahn et al., 2018) | ×4 | 32.13 | 0.8937 | 28.60 | 0.7806 | 27.58 | 0.7349 | 26.07 | 0.7837 | 30.46 | 0.9083 |
| IMDN (Hui et al., 2019) | ×4 | 32.21 | 0.8948 | 28.58 | 0.7811 | 27.56 | 0.7353 | 26.04 | 0.7838 | 30.45 | 0.9075 |
| LatticeNet (Luo et al., 2022) | ×4 | 32.18 | 0.8943 | 28.61 | 0.7812 | 27.57 | 0.7355 | 26.14 | 0.7844 | N/A | N/A |
| ASSLN (Zhang et al., 2021) | ×4 | 32.29 | 0.8964 | 28.69 | 0.7844 | 27.66 | 0.7384 | 26.27 | 0.7907 | 30.84 | 0.9119 |
| FMP (ours) | ×4 | 32.34 | 0.8979 | 28.71 | 0.7878 | 27.67 | 0.7425 | 26.35 | 0.7954 | 30.90 | 0.9132 |

Table 1: PSNR/SSIM comparisons. Best and second best results are colored with red and blue.

**Model Complexity**. In Tab. 2, we provide model complexity comparison. Several lightweight SR models (*e.g.*, SRCNN and FS-RCNN) achieve a very small number of parameters and FLOPs, yet have limited performance. Compared with recent leading works (*e.g.*, IMDN, LatticeNet, and ASSLN), our FMP has comparable parameter numbers and FLOPs. FMP operates fewer FLOPs than most compared methods. When considering Tabs. 1 and 2 together, our FMP achieves a good trade-off between performance and model complexity.

| Method | ×2 Params | ×2 FLOPs | ×3 Params | ×3 FLOPs | ×4 Params | ×4 FLOPs |
|---|---|---|---|---|---|---|
| SRCNN | 57K | 52.7G | 57K | 52.7G | 57K | 52.7G |
| FSRCNN | 12K | 6.0G | 12K | 5.0G | 12K | 4.6G |
| VDSR | 665K | 612.6G | 665K | 612.6G | 665K | 612.6G |
| DRCN | 1,774K | 17,974.3G | 1,774K | 17,974.3G | 1,774K | 17,974.3G |
| LapSRN | 813K | 29.9G | N/A | N/A | 813K | 149.4G |
| DRRN | 297K | 6,796.9G | 297K | 6,796.9G | 297K | 6,796.9G |
| MemNet | 677K | 2,662.4G | 677K | 2,662.4G | 677K | 2,662.4G |
| CARN | 1,592K | 222.8G | 1,592K | 118.8G | 1,592K | 90.9G |
| IMDN | 694K | 158.8G | 703K | 71.5G | 715K | 40.9G |
| LatticeNet | 756K | 169.5G | 765K | 76.3G | 777K | 43.6G |
| ASSLN | 692K | 159.1G | 698K | 71.2G | 708K | 40.6G |
| FMP (ours) | 694K | 153.7G | 684K | 67.3G | 704K | 39.0G |

Table 2: Model size and FLOPs comparisons.

**Visual Results**. We further provide visual results (×4) in Fig. 5. In img_083, we can observe that most compared methods either hardly reconstruct structural details with proper directions or suffer from blurring artifacts. In contrast, our FMP can recover more structural details and better alleviate the blurring artifacts. Other similar observations can also be easily found. These visual comparisons are consistent with the trend in quantitative results, indicating the superiority of our method.

## 4.3 ABLATION STUDY

We train all models from scratch for ablation study. The input size is 3×64×64 for FLOPs calculation. Training process will stop if meets convergence or reach the maximum iterations 300K.

**Effectiveness of LSRB**. To show the effectiveness of the newly designed LSRB, we first compare it with RLFN (Kong et al., 2022). To keep similar model complexity, we configure LSRB-6-48, which consists

| Method | Params | FLOPs | Inference Time (ms) Urban100 | Inference Time (ms) DIV2K | Urban100 PSNR | Urban100 SSIM |
|---|---|---|---|---|---|---|
| RLFN | 0.32 M | 1.23G | 19 | 34 | 25.54 | 0.7675 |
| LSRB-6-48 | 0.35 M | 1.31G | 12 | 23 | 25.62 | 0.7700 |

Table 3: Quantitative results (×4) of different lightweight SR models. Inference time is tested with an RTX 3090 GPU.

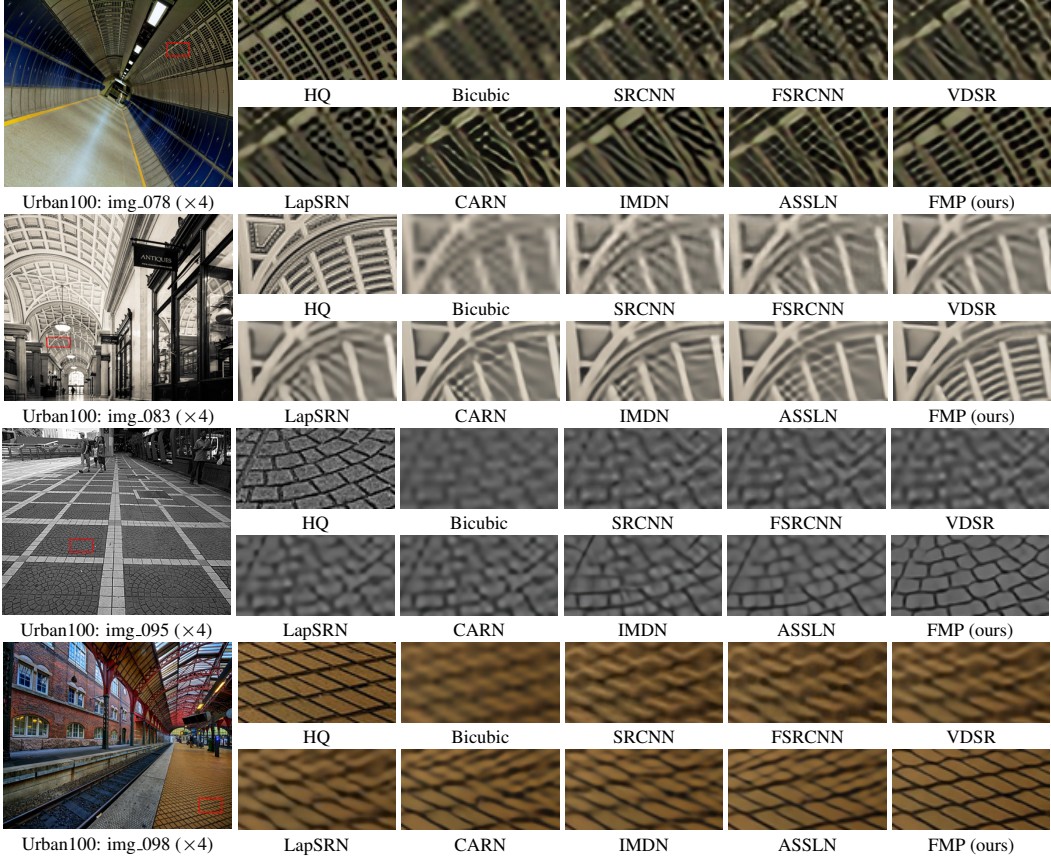

Figure 5: Visual comparison (×4) with lightweight SR networks on Urban100 dataset.

of 6 basic blocks (RBs) with 48 channels for each Conv layer. We report inference time on Urban100 and DIV2K validation and test data. In Tab. 3, our LSRB achieves much faster better performance than RLFN at the cost of slightly more parameters and FLOPs. This observation indicates that LSRB achieves a good trade-off among inference time, model complexity (*i.e.*, parameters and FLOPs), and performance. It is promising to further reduce its redundant parameters with our proposed FMP.

**Flexible vs. Channel Pruning**. We then compare with channel pruning methods in image SR. In the pruning stage, we do not use pretrained models, which are needed in ASSL (Zhang et al., 2021) and SRP (Zhang et al., 2022b). Consequently, we compare with DHP (Li et al., 2020a), which only conducts channel pruning without pretraining. In Tab. 4, we use EDSR-8-128 as

| Prune Ratio (%) | Method | Set5 | Set14 | B100 | Urban100 |
|---|---|---|---|---|---|
| 60 | DHP | 31.99 | 28.52 | 27.53 | 25.92 |
| | FMP (ours) | 32.16 | 28.60 | 27.55 | 25.96 |
| 40 | DHP | 32.01 | 28.49 | 27.52 | 25.86 |
| | FMP (ours) | 32.08 | 28.58 | 27.53 | 25.91 |
| 20 | DHP | 31.94 | 28.42 | 27.47 | 25.69 |
| | FMP (ours) | 31.97 | 28.51 | 27.47 | 25.78 |

Table 4: Flexible vs. channel pruning in EDSR-8-128.

the image SR backbone, which has been used in DHP (Li et al., 2020a) and consists of 8 RBs with 128 channels for each convolutional (Conv) layer. By additionally pruning kernel weights, FMP performs better than DHP across different cases. It indicates that flexibly pruning both network channels and weights achieves further improvements.

**Pruning Method**. As mentioned in Sec. 3.4, sparsity regularization can be applied to the channel vectors and weight indicators. The optimization of channel vectors has been sufficiently studied in DHP (Li et al., 2020a). Thus, we just use the default $L_1$ norm regularization for channel vector optimization. By contrast, we study the sparsity regularization on the weight indicators thoroughly in this paper. Specifically, we study three regularization terms in Eq. 7 including $L_1$ norm, $L_2$ norm, and weight decay regularization in Tab. 5. We find that $L_1$ norm and weight decay could perform better than $L_2$ norm. Meanwhile, the $L_1$ norm performs faster convergence than weight decay. Therefore, we choose $L_1$ norm for weight indicator optimization.

| Method | Total Ratio (%) | | Channel Prune Ratio (%) | | Weight Prune Ratio (%) | | PSNR (dB) of EDSR-8-128 + FMP | | | | |
|---|---|---|---|---|---|---|---|---|---|---|---|
| | FLOPs | Params | FLOPs | Params | FLOPs | Params | Set5 | Set14 | B100 | Urban100 | Manga109 |
| $L_1$ norm | 61.98 | 57.32 | 24.89 | 25.51 | 13.12 | 17.17 | 32.01 | 28.52 | 27.51 | 25.90 | 30.10 |
| $L_2$ norm | 61.87 | 58.05 | 8.90 | 10.16 | 29.24 | 31.78 | 31.97 | 28.49 | 27.49 | 25.81 | 30.01 |
| Weight Decay | 61.95 | 59.98 | 31.11 | 31.49 | 6.94 | 8.53 | 32.03 | 28.53 | 27.52 | 25.90 | 30.10 |

Table 5: Weight pruning methods in FMP for image SR (×4). We apply FMP to EDSR-8-128.

| Metric | Criteria | Total Ratio (%) | | Channel Prune Ratio (%) | | Weight Prune Ratio (%) | | PSNR (dB) of EDSR-8-128 + FMP | | | | |
|---|---|---|---|---|---|---|---|---|---|---|---|---|
| | | FLOPs | Params | FLOPs | Params | FLOPs | Params | Set5 | Set14 | B100 | Urban100 | Manga109 |
| Params | Channel | 72.88 | 70.61 | 18.34 | 18.51 | 8.78 | 10.88 | 32.00 | 28.51 | 28.51 | 25.94 | 30.05 |
| | Weight | 48.55 | 41.49 | 33.24 | 33.39 | 18.21 | 25.12 | 31.90 | 28.45 | 27.46 | 25.76 | 29.87 |
| | Total Fixed | 62.23 | 55.59 | 19.65 | 19.78 | 18.11 | 24.63 | 31.98 | 28.50 | 27.51 | 25.85 | 30.04 |
| | Total | 61.95 | 59.98 | 31.11 | 31.49 | 6.94 | 8.53 | 32.03 | 28.53 | 27.52 | 25.90 | 30.10 |
| FLOPs | Channel | 72.93 | 70.70 | 18.34 | 18.51 | 8.73 | 10.78 | 32.04 | 28.56 | 27.53 | 25.96 | 30.15 |
| | Weight | 59.19 | 54.39 | 27.19 | 27.60 | 13.63 | 18.01 | 31.97 | 28.49 | 27.50 | 25.90 | 30.02 |
| | Total Fixed | 67.46 | 62.70 | 18.64 | 18.95 | 13.90 | 18.35 | 31.97 | 28.54 | 27.53 | 25.91 | 30.07 |
| | Total | 65.39 | 61.57 | 23.15 | 23.60 | 11.45 | 14.83 | 32.04 | 28.55 | 27.53 | 25.94 | 30.11 |

Table 6: Convergence criteria in FMP for image SR (×4). We apply FMP to EDSR-8-128.

## 4.4 CONVERGENCE CRITERIA

As shown in Sec. 3.4, the convergence criteria needs to be defined during the optimization of the pruning algorithm. In the paper, we define the pruning ratio $\gamma_C$ and $\gamma_W$ in terms of either the number of parameters or FLOPs, depending on which metric we want to optimize for. Both structured pruning and unstructured pruning are conducted during the optimization. In addition, we defined four convergence criteria: **(1)** Channel: the pruning algorithm converges if the channel pruning ratio $\gamma_C$ is achieved. **(2)** Weight: the pruning algorithm converges if the channel pruning ratio $\gamma_W$ is achieved. **(3)** Total Fixed: both the pruning ratio $\gamma_C$ and $\gamma_W$ should be met individually. **(4)** Total: the joint pruning ratio $\gamma_C + \gamma_W$ is achieved. The percentage of weight pruning and channel pruning is determined automatically. We provide results in Tab. 6. We can learn that pruning channel and weight jointly (*i.e.*, Total Fixed and Total cases) reduces more parameters and obtains comparable performance as channel pruning alone.

## 4.5 DIFFERENT MODEL COMPRESSION METHODS

To further show effectiveness of flexible network pruning method, we compare FMP with representative model compression techniques for image SR. Specifically, we compare with neural architecture search (NAS) based methods (*i.e.*, MoreMNAS-A (Chu et al., 2019b) and FALSR-A (Chu et al., 2019a)), knowledge distillation (KD) based methods

| Method | Type | Params | FLOPs | Set5 | B100 |
|---|---|---|---|---|---|
| MoreMNAS-A | NAS | 1,039K | 238.6G | 37.63 | 31.95 |
| FALSR-A | NAS | 1,021K | 234.7G | 37.82 | 32.12 |
| CARN+KD | KD | 1,592K | 222.8G | 37.82 | 32.08 |
| ASSLN | C Prune | 692K | 159.1G | 38.12 | 32.27 |
| FMP (ours) | C+W Prune | 694K | 153.7G | 38.17 | 32.32 |

Table 7: Parameters, FLOPs, and PSNR comparisons (×2). 'C' and 'W' denote channel and weight.

(*i.e.*, CARN+KD (Lee et al., 2020)), and channel pruning based method (*i.e.*, ASSLN (Zhang et al., 2021)). We provide quantitative results in Tab. 7. Our FMP obtains the best performance (see Tab. 1) with comparable parameters and FLOPs as others. We do not have to search lots of architectures or train a teacher network as NAS and KD based methods do. ASSLN prunes channels from a pretrained model. With our proposed flexible network pruning strategy, we can prune channels and weights jointly without pretrained models, being more flexible than ASSLN.

## 5 CONCLUSION

In this work, we design a lightweight SR baseline (LSRB), which runs fast yet obtains comparable performance as other lightweight models. We then propose a flexible meta pruning (FMP) technique to prune network channels and weights simultaneously. Specifically, we introduce a hypernetwork, taking channel vectors and weight indicators as inputs. The hypernetwork outputs serve as the network parameters for the SR backbone. Consequently, for each network layer, we conduct structured pruning with channel vectors, controlling the output and input channels. Besides, we conduct unstructured pruning with weight indicators to influence the sparsity of kernel weights, resulting in flexible pruning. During pruning, both channel vectors and weight indicators are regularized by sparsity and are optimized with proximal gradient and SGD respectively. We conduct extensive experiments to investigate effect of key factors in our FMP, such as convergence criteria and pruning method. Our FMP also achieves superior performance gains over recent leading methods.

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
