# SUPPLEMENTARY MATERIALS: LIGHTWEIGHT IMAGE SUPER-RESOLUTION VIA FLEXIBLE META PRUNING

## 1  MORE DISCUSSIONS

### 1.1  MORE CLARIFICATIONS ABOUT NOVELTY

The idea of combining unstructured pruning and structured pruning for image SR is straightforward. However, how to design an algorithm to achieve flexible pruning is still worthy of investigation. Our FMP could automatically allocate parameters and computation budgets for unstructured and structured pruning.

The introduced technique such as weight indicator extends the usage of hypernetworks from channel pruning to a wider scope of network (weight and channel) pruning.

### 1.2  MORE DISCUSSIONS ABOUT INFERENCE TIME

Our FMP also reduces network redundancy and the resulting models are more friendly for on-device storage and transmission for inference usage. The inference time can be further improved by AI accelerators, since the computation is reduced. But, the hardware design related to network pruning is out of the scope of this paper.

### 1.3  DIFFERENCES BETWEEN FMP AND ASSLN (ZHANG ET AL., 2021)

**(1)** ASSLN needs a large pre-trained model for pruning, while FMP **does not**. **(2)** ASSLN only prunes channels. FMP prunes both channels and weights flexibly. **(3)** FMP further reduces network redundancy, enhances network representation ability, and obtains better reconstruction results.

## 2  EXPERIMENTAL RESULTS

### 2.1  ARM ESRB WITH FMP

We give more details about applying our flexible meta pruning (FMP) to the designed efficient super-resolution baseline (ESRB). We use ESRB-6-256 as the original model and prune it to the target lightweight one (denoted as FMP for simplicity). ESRB-6-256 consists of 6 basic blocks and 256 channels in the convolutional layer. It should be noted that we do not pretrain ESRB-6-256. This is different from ASSLN (Zhang et al., 2021), which needs a pretained model.

We provide the parameters (*i.e.*, Params), FLOPs, and prune ratio in Tab. 1. We prune the original large model ESRB-6-256 by large prune ratio. Namely, the parameter prune ratios are 91.52%, 91.67%, and 91.48% with respect to ×2, ×2, and ×4. The FLOPs prune ratios follow a similar trend. Our FMP can flexibly prune large models by large prune ratios with more efficient parameters.

### 2.2  MORE ANALYSES ABOUT CONVERGENCE CRITERIA

We provide results in Tab. 3 to investigate the convergence criteria, which needs to be defined during the optimization. We define the pruning ratio $\gamma_C$ and $\gamma_W$ in terms of either the number of parameter (denoted as Params) or FLOPs, depending on which metric we want to optimize for. Both structured pruning (*i.e.*, channel pruning) and unstructured pruning (*i.e.*, weight pruning) are conducted during the optimization. In addition, we defined four convergence criteria: **(1)** Channel: the pruning algorithm converges if the channel pruning ratio $\gamma_C$ is achieved. **(2)** Weight: the pruning algorithm

| Scale | ESRB-6-256 | | FMP | | Prune Ratio (%) | |
|---|---|---|---|---|---|---|
| | Params | FLOPs | Params | FLOPs | Params | FLOPs |
| ×2 | 8,180K | 1,877.8G | 694K | 153.7G | 91.52 | 91.81 |
| ×3 | 8,215K | 836.8G | 684K | 67.3G | 91.67 | 91.96 |
| ×4 | 8,264K | 474.2G | 704K | 39.0G | 91.48 | 91.78 |

Table 1: Model size, FLOPs, and prune ratio before and after pruning. We set output size as $3×1280×720$ to calculate FLOPs.

| Method | ×2 | | ×3 | | ×4 | |
|---|---|---|---|---|---|---|
| | Params | FLOPs | Params | FLOPs | Params | FLOPs |
| SRCNN (Dong et al., 2014) | 57K | 52.7G | 57K | 52.7G | 57K | 52.7G |
| FSRCNN (Dong et al., 2016) | 12K | 6.0G | 12K | 5.0G | 12K | 4.6G |
| VDSR (Kim et al., 2016a) | 665K | 612.6G | 665K | 612.6G | 665K | 612.6G |
| DRCN (Kim et al., 2016b) | 1,774K | 17,974.3G | 1,774K | 17,974.3G | 1,774K | 17,974.3G |
| LapSRN (Lai et al., 2017) | 813K | 29.9G | N/A | N/A | 813K | 149.4G |
| DRRN (Tai et al., 2017a) | 297K | 6,796.9G | 297K | 6,796.9G | 297K | 6,796.9G |
| MemNet (Tai et al., 2017b) | 677K | 2,662.4G | 677K | 2,662.4G | 677K | 2,662.4G |
| SelNet (Choi & Kim, 2017) | 974K | 225.7G | 1,159K | 120.0G | 1,417K | 83.1G |
| CARN (Ahn et al., 2018) | 1,592K | 222.8G | 1,592K | 118.8G | 1,592K | 90.9G |
| BSRN (Choi et al., 2018) | 594K | 1666.7G | 779K | 761.1G | 742K | 451.8G |
| IMDN (Hui et al., 2019) | 694K | 158.8G | 703K | 71.5G | 715K | 40.9G |
| LatticeNet (Luo et al., 2022) | 756K | 169.5G | 765K | 76.3G | 777K | 43.6G |
| ASSLN (Zhang et al., 2021) | 692K | 159.1G | 698K | 71.2G | 708K | 40.6G |
| FMP (ours) | 694K | 153.7G | 684K | 67.3G | 704K | 39.0G |

Table 2: Model size and FLOPs comparisons.

converges if the channel pruning ratio $\gamma_W$ is achieved. **(3)** Total Fixed: both the pruning ratio $\gamma_C$ and $\gamma_W$ should be met individually. **(4)** Total: the joint pruning ratio $\gamma_C + \gamma_W$ is achieved. The percentage of weight pruning and channel pruning is determined automatically.

In Tab. 3, we use Params and FLOPs as metrics in the pruning process. For each metric, we further use four criteria: 'Channel', 'Weight', 'Total Fixed', and 'Total' for convergence. The term 'Total Ratio (%)' means the remaining ratio in terms of FLOPs or Params after pruning. The terms 'Channel Prune Ratio (%)' and 'Weight Prune Ratio (%)' mean the amount ratio pruned with respect to channel and weight. We can learn that pruning channel and weight jointly (*i.e.*, Total Fixed and Total cases) reduces more parameters and obtains comparable performance as channel pruning alone. We take 'Total' in the experiments.

## 2.3 Main Comparisons

We compare our lightweight network FMP with representative lightweight SR networks: SRCNN (Dong et al., 2014), FSRCNN (Dong et al., 2016), VDSR (Kim et al., 2016a), DRCN (Kim et al., 2016b), CNF (Ren et al., 2017), LapSRN (Lai et al., 2017), DRRN (Tai et al., 2017a), MemNet (Tai et al., 2017b), SelNet (Choi & Kim, 2017), CARN (Ahn et al., 2018), BSRN (Choi et al., 2018), IMDN (Hui et al., 2019), LatticeNet (Luo et al., 2022), and ASSLN (Zhang et al., 2021).

**Quantitative Results**. Table 4 shows PSNR/SSIM comparisons for ×2, ×3, and ×4 SR. ASSLN (Zhang et al., 2021) ranks the second best. When compared to all other methods, our FMP performs the best on all the datasets and scaling factors. Specifically, let's take ×2 SR as an example. FMP obtains about 0.30 dB on Urban100 PSNR gains over ASSLN. These comparisons show the effectiveness of FMP, which prunes the network channels and weights flexibly.

**Visual Results**. We further provide visual comparisons (×4) in Figs. 1, 2, 3, and 4 for challenging cases. For example, in img_072, we can observe that most of the compared methods suffer from heavy blurring artifacts in the challenging cases (*e.g.*., img_033 and img_059). They can hardly reconstruct structural details with proper directions (*e.g.*., img_061 and img_073). While, our FMP can better alleviate the blurring artifacts and recover more structural and texture details (*e.g.*, img_091). Similar observations can be found in other images. These visual comparisons are consistent with the quantitative comparisons, demonstrating the effectiveness of our method.

| Metric | Criteria | Total Ratio (%) | | Channel Prune Ratio (%) | | Weight Prune Ratio (%) | | PSNR (dB) of EDSR-8-128 + FMP | | | | |
|--------|----------|-------|--------|-------|--------|-------|--------|------|-------|------|---------|----------|
| | | FLOPs | Params | FLOPs | Params | FLOPs | Params | Set5 | Set14 | B100 | Urban100 | Manga109 |
| Params | Channel | 72.88 | 70.61 | 18.34 | 18.51 | 8.78 | 10.88 | 32.00 | 28.51 | 28.51 | 25.94 | 30.05 |
| | Weight | 48.55 | 41.49 | 33.24 | 33.39 | 18.21 | 25.12 | 31.90 | 28.45 | 27.46 | 25.76 | 29.87 |
| | Total Fixed | 62.23 | 55.59 | 19.65 | 19.78 | 18.11 | 24.63 | 31.98 | 28.50 | 27.51 | 25.85 | 30.04 |
| | Total | 61.95 | 59.98 | 31.11 | 31.49 | 6.94 | 8.53 | 32.03 | 28.53 | 27.52 | 25.90 | 30.10 |
| FLOPs | Channel | 72.93 | 70.70 | 18.34 | 18.51 | 8.73 | 10.78 | 32.04 | 28.56 | 27.53 | 25.96 | 30.15 |
| | Weight | 59.19 | 54.39 | 27.19 | 27.60 | 13.63 | 18.01 | 31.97 | 28.49 | 27.50 | 25.90 | 30.02 |
| | Total Fixed | 67.46 | 62.70 | 18.64 | 18.95 | 13.90 | 18.35 | 31.97 | 28.54 | 27.53 | 25.91 | 30.07 |
| | Total | 65.39 | 61.57 | 23.15 | 23.60 | 11.45 | 14.83 | 32.04 | 28.55 | 27.53 | 25.94 | 30.11 |

Table 3: Convergence criteria in FMP for image SR (×4). We apply FMP to EDSR-8-128.

**Model Complexity**. We provide parameter number and FLOPs comparison in Tab. 2. Although some previous lightweight SR models (*e.g.*, SRCNN and FSRCNN) cost very small model sizes and FLOPs, they have limited performance. Compared with recent popular works (*e.g.*, MemNet, CARN, IMDN, LatticeNet, and ASSLN), our FMP has the least parameter number. The FLOPs comparison also follow similar trend. Our FMP operates least F LOPs than most recent methods. When we consider Tabs. 4 and 2 together, we find that our FMP achieves a better trade-off between performance and resource consumption and reduces parameters and operations efficiently.

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

| Method | Scale | Set5 | | Set14 | | B100 | | Urban100 | | Manga109 | |
|---|---|---|---|---|---|---|---|---|---|---|---|
| | | PSNR | SSIM | PSNR | SSIM | PSNR | SSIM | PSNR | SSIM | PSNR | SSIM |
| SRCNN | ×2 | 36.66 | 0.9542 | 32.42 | 0.9063 | 31.36 | 0.8879 | 29.50 | 0.8946 | 35.60 | 0.9663 |
| FSRCNN | ×2 | 37.00 | 0.9558 | 32.63 | 0.9088 | 31.53 | 0.8920 | 29.88 | 0.9020 | 36.67 | 0.9710 |
| VDSR | ×2 | 37.53 | 0.9587 | 33.03 | 0.9124 | 31.90 | 0.8960 | 30.76 | 0.9140 | 37.22 | 0.9750 |
| DRCN | ×2 | 37.63 | 0.9588 | 33.04 | 0.9118 | 31.85 | 0.8942 | 30.75 | 0.9133 | 37.63 | 0.9740 |
| CNF | ×2 | 37.66 | 0.9590 | 33.38 | 0.9136 | 31.91 | 0.8962 | N/A | N/A | N/A | N/A |
| LapSRN | ×2 | 37.52 | 0.9590 | 33.08 | 0.9130 | 31.80 | 0.8950 | 30.41 | 0.9100 | 37.27 | 0.9740 |
| DRRN | ×2 | 37.74 | 0.9591 | 33.23 | 0.9136 | 32.05 | 0.8973 | 31.23 | 0.9188 | 37.92 | 0.9760 |
| MemNet | ×2 | 37.78 | 0.9597 | 33.28 | 0.9142 | 32.08 | 0.8978 | 31.31 | 0.9195 | 37.72 | 0.9740 |
| SelNet | ×2 | 37.89 | 0.9598 | 33.61 | 0.9160 | 32.08 | 0.8984 | N/A | N/A | N/A | N/A |
| CARN | ×2 | 37.76 | 0.9590 | 33.52 | 0.9166 | 32.09 | 0.8978 | 31.92 | 0.9256 | 38.36 | 0.9764 |
| BSRN | ×2 | 37.78 | 0.9591 | 33.43 | 0.9155 | 32.11 | 0.8983 | 31.92 | 0.9261 | N/A | N/A |
| FALSR-A | ×2 | 37.82 | 0.9595 | 33.55 | 0.9168 | 32.12 | 0.8987 | 31.93 | 0.9256 | N/A | N/A |
| IMDN | ×2 | 38.00 | 0.9605 | 33.63 | 0.9177 | 32.19 | 0.8996 | 32.17 | 0.9283 | 38.87 | 0.9773 |
| LatticeNet | ×2 | 38.06 | 0.9607 | 33.70 | 0.9187 | 32.20 | 0.8999 | 32.25 | 0.9288 | N/A | N/A |
| ASSLN | ×2 | 38.12 | 0.9608 | 33.77 | 0.9194 | 32.27 | 0.9007 | 32.41 | 0.9309 | 39.12 | 0.9781 |
| FMP (ours) | ×2 | 38.17 | 0.9615 | 33.81 | 0.9215 | 32.32 | 0.9022 | 32.71 | 0.9360 | 39.17 | 0.9783 |
| SRCNN | ×3 | 32.75 | 0.9090 | 29.28 | 0.8209 | 28.41 | 0.7863 | 26.24 | 0.7989 | 30.48 | 0.9117 |
| FSRCNN | ×3 | 33.16 | 0.9140 | 29.43 | 0.8242 | 28.53 | 0.7910 | 26.43 | 0.8080 | 31.10 | 0.9210 |
| VDSR | ×3 | 33.66 | 0.9213 | 29.77 | 0.8314 | 28.82 | 0.7976 | 27.14 | 0.8279 | 32.01 | 0.9340 |
| DRCN | ×3 | 33.82 | 0.9226 | 29.76 | 0.8311 | 28.80 | 0.7963 | 27.15 | 0.8276 | 32.31 | 0.9360 |
| DRRN | ×3 | 34.03 | 0.9244 | 29.96 | 0.8349 | 28.95 | 0.8004 | 27.53 | 0.8378 | 32.74 | 0.9390 |
| MemNet | ×3 | 34.09 | 0.9248 | 30.00 | 0.8350 | 28.96 | 0.8001 | 27.56 | 0.8376 | 32.51 | 0.9369 |
| SelNet | ×3 | 34.27 | 0.9257 | 30.30 | 0.8399 | 28.97 | 0.8025 | N/A | N/A | N/A | N/A |
| CARN | ×3 | 34.29 | 0.9255 | 30.29 | 0.8407 | 29.06 | 0.8034 | 28.06 | 0.8493 | 33.50 | 0.9539 |
| IMDN | ×3 | 34.36 | 0.9270 | 30.32 | 0.8417 | 29.09 | 0.8046 | 28.17 | 0.8519 | 33.61 | 0.9444 |
| BSRN | ×3 | 34.32 | 0.9255 | 30.25 | 0.8404 | 29.07 | 0.8039 | 28.04 | 0.8497 | N/A | N/A |
| LatticeNet | ×3 | 34.40 | 0.9272 | 30.32 | 0.8416 | 29.10 | 0.8049 | 28.19 | 0.8513 | N/A | N/A |
| ASSLN | ×3 | 34.51 | 0.9280 | 30.45 | 0.8439 | 29.19 | 0.8069 | 28.35 | 0.8562 | 34.00 | 0.9468 |
| FMP (ours) | ×3 | 34.55 | 0.9291 | 30.48 | 0.8456 | 29.20 | 0.8101 | 28.40 | 0.8597 | 34.06 | 0.9473 |
| SRCNN | ×4 | 30.48 | 0.8628 | 27.49 | 0.7503 | 26.90 | 0.7101 | 24.52 | 0.7221 | 27.58 | 0.8555 |
| FSRCNN | ×4 | 30.71 | 0.8657 | 27.59 | 0.7535 | 26.98 | 0.7150 | 24.62 | 0.7280 | 27.90 | 0.8610 |
| VDSR | ×4 | 31.35 | 0.8838 | 28.01 | 0.7674 | 27.29 | 0.7251 | 25.18 | 0.7524 | 28.83 | 0.8870 |
| DRCN | ×4 | 31.53 | 0.8854 | 28.02 | 0.7670 | 27.23 | 0.7233 | 25.14 | 0.7510 | 28.98 | 0.8870 |
| CNF | ×4 | 31.55 | 0.8856 | 28.15 | 0.7680 | 27.32 | 0.7253 | N/A | N/A | N/A | N/A |
| LapSRN | ×4 | 31.54 | 0.8850 | 28.19 | 0.7720 | 27.32 | 0.7280 | 25.21 | 0.7560 | 29.09 | 0.8900 |
| DRRN | ×4 | 31.68 | 0.8888 | 28.21 | 0.7720 | 27.38 | 0.7284 | 25.44 | 0.7638 | 29.46 | 0.8960 |
| MemNet | ×4 | 31.74 | 0.8893 | 28.26 | 0.7723 | 27.40 | 0.7281 | 25.50 | 0.7630 | 29.42 | 0.8942 |
| SelNet | ×4 | 32.00 | 0.8931 | 28.49 | 0.7783 | 27.44 | 0.7325 | N/A | N/A | N/A | N/A |
| CARN | ×4 | 32.13 | 0.8937 | 28.60 | 0.7806 | 27.58 | 0.7349 | 26.07 | 0.7837 | 30.46 | 0.9083 |
| BSRN | ×4 | 32.14 | 0.8937 | 28.56 | 0.7803 | 27.57 | 0.7353 | 26.03 | 0.7835 | N/A | N/A |
| IMDN | ×4 | 32.21 | 0.8948 | 28.58 | 0.7811 | 27.56 | 0.7353 | 26.04 | 0.7838 | 30.45 | 0.9075 |
| LatticeNet | ×4 | 32.18 | 0.8943 | 28.61 | 0.7812 | 27.57 | 0.7355 | 26.14 | 0.7844 | N/A | N/A |
| ASSLN | ×4 | 32.29 | 0.8964 | 28.69 | 0.7844 | 27.66 | 0.7384 | 26.27 | 0.7907 | 30.84 | 0.9119 |
| FMP (ours) | ×4 | 32.34 | 0.8979 | 28.71 | 0.7878 | 27.67 | 0.7425 | 26.35 | 0.7954 | 30.90 | 0.9132 |

Table 4: PSNR/SSIM comparisons about lightweight image SR. Best and second best results are colored with red and blue.

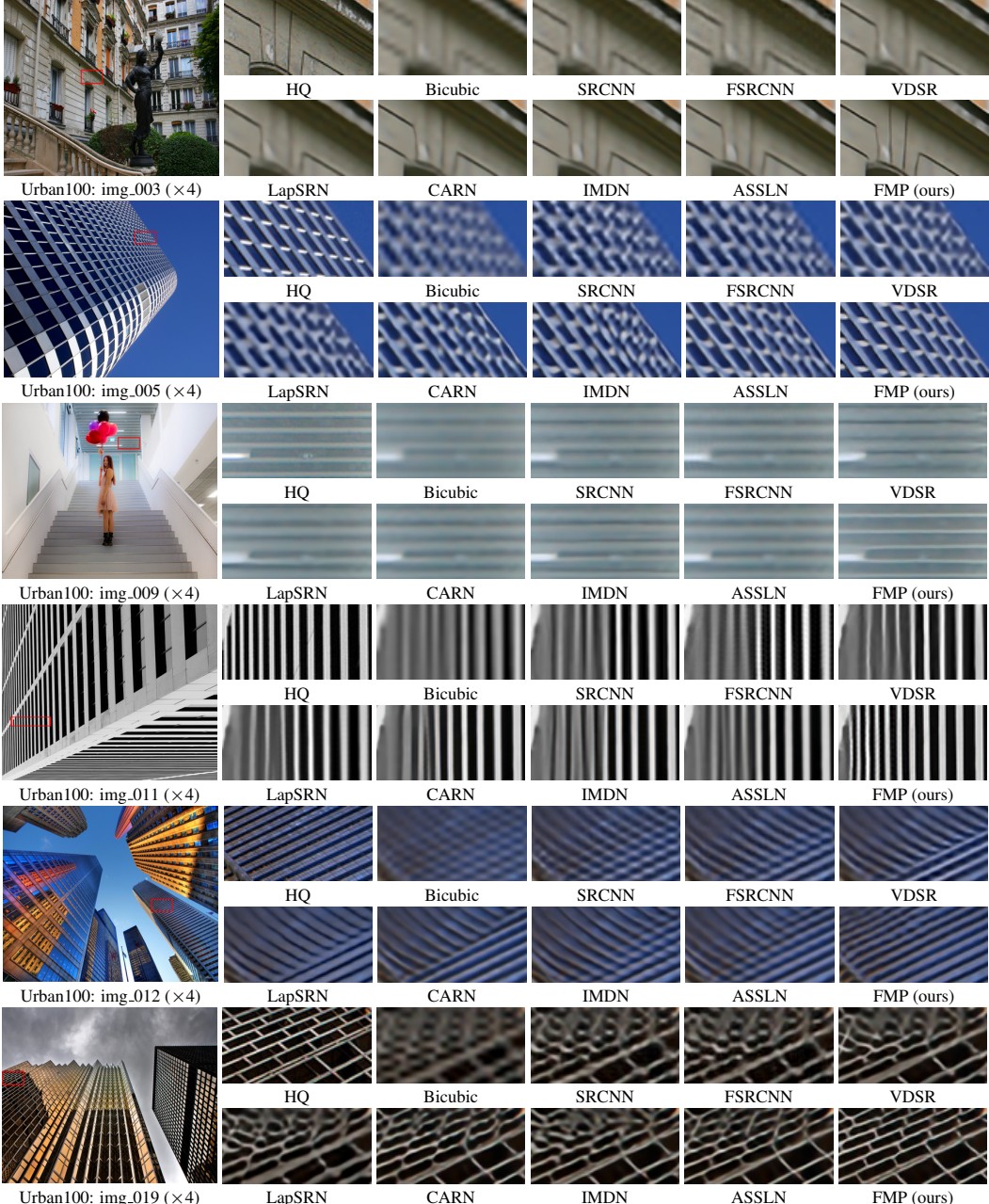

Figure 1: Visual comparison (×4) with lightweight SR networks on Urban100 dataset.

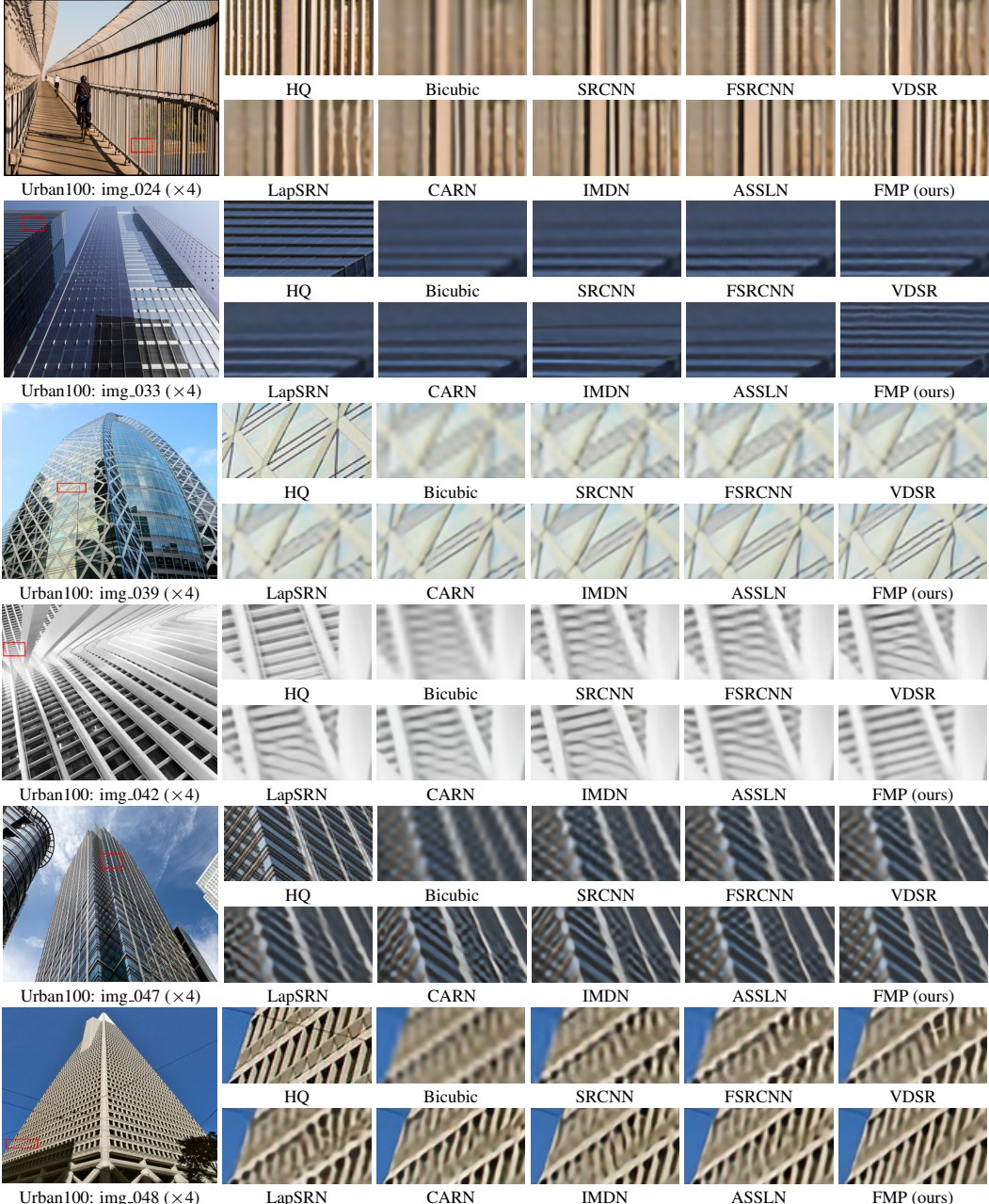

Figure 2: Visual comparison (×4) with lightweight SR networks on Urban100 dataset.

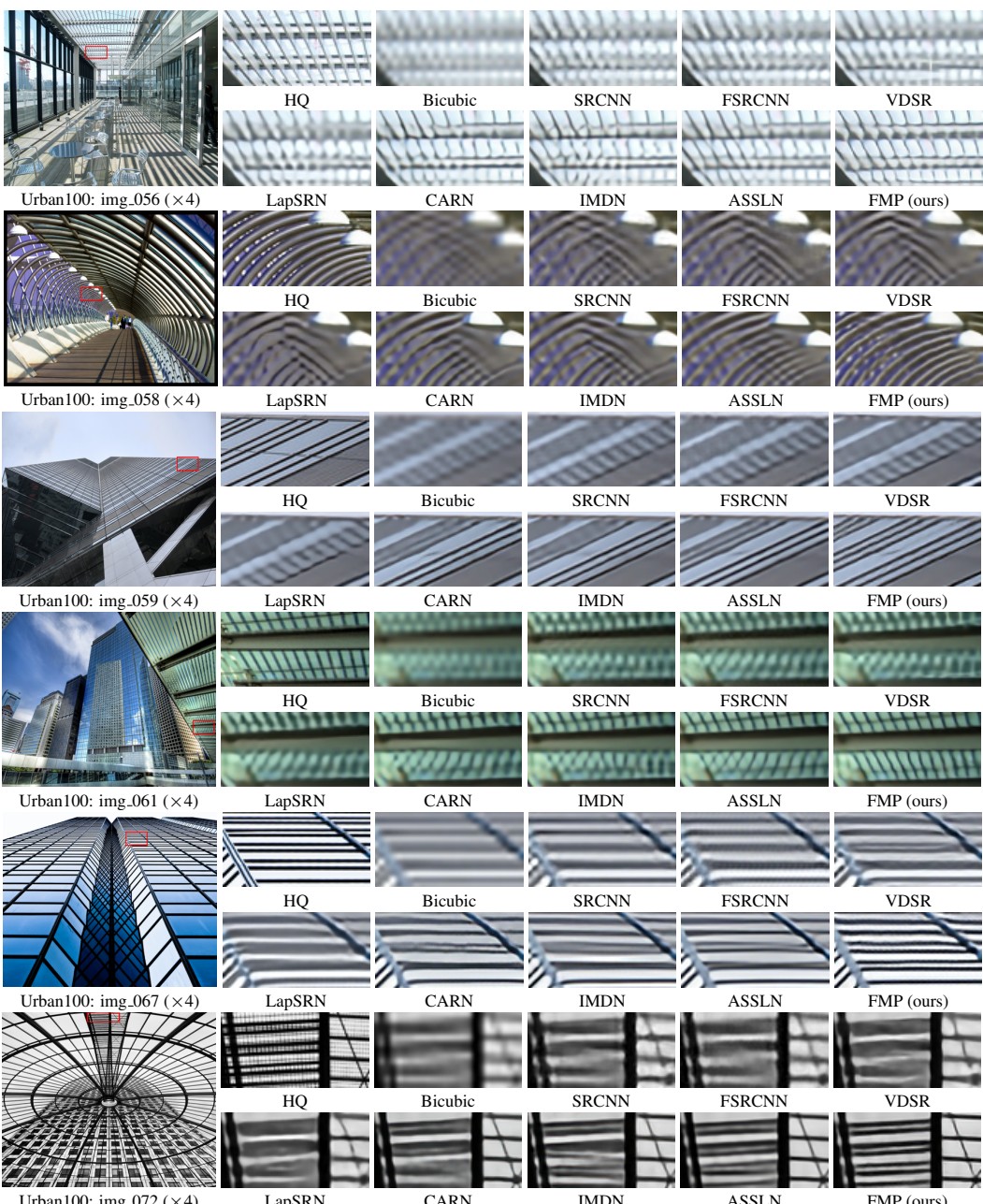

Figure 3: Visual comparison (×4) with lightweight SR networks on Urban100 dataset.

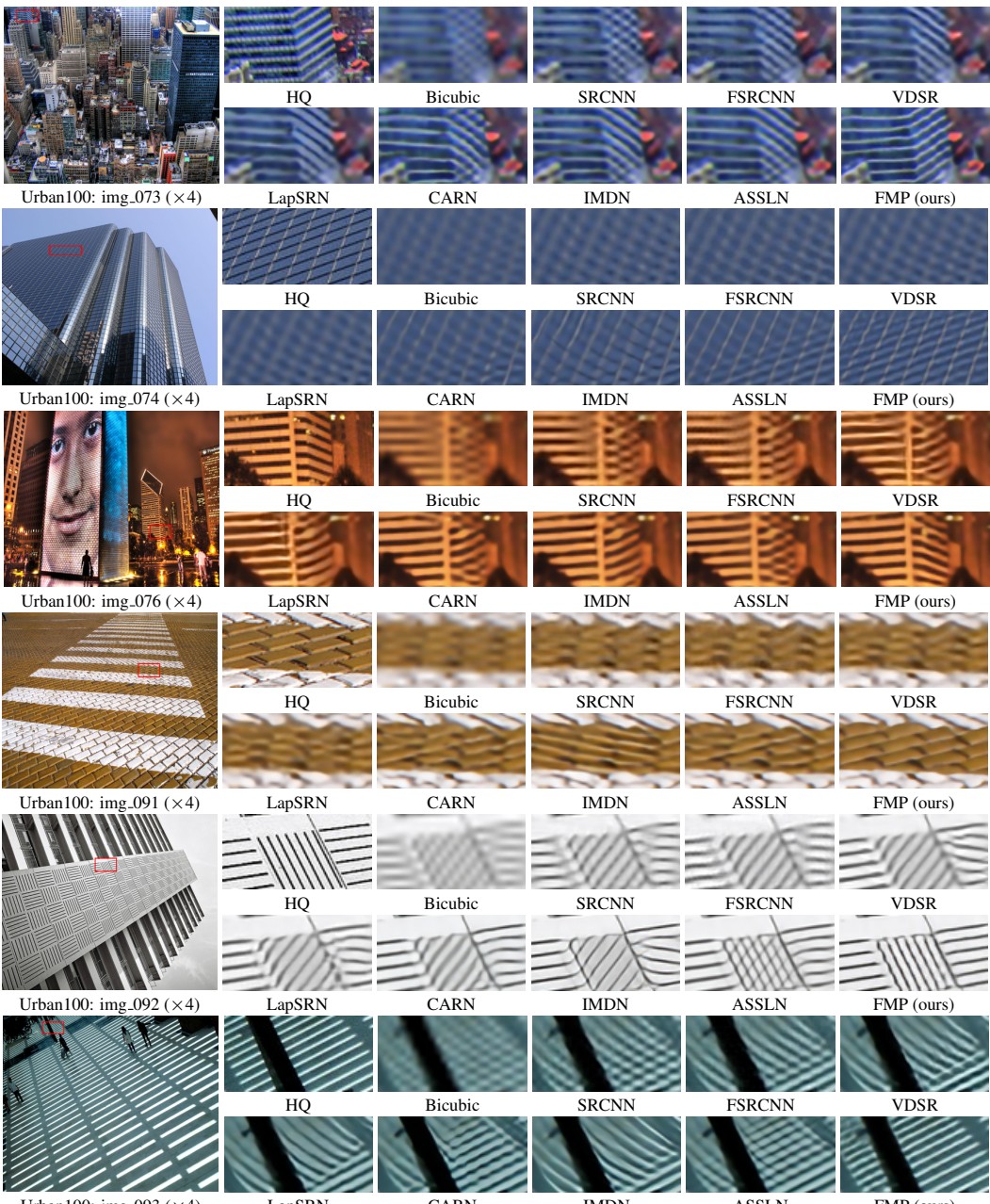

Figure 4: Visual comparison (×4) with lightweight SR networks on Urban100 dataset.