# OpenReview forum: "Lightweight Image Super-Resolution via Flexible Meta Pruning"
_ICLR.cc/2024/Conference — Submitted to ICLR 2024_

### Official Review · Reviewer_pKAh · 2023-10-26

**Soundness:** 3 good
**Presentation:** 2 fair
**Contribution:** 3 good
**Rating:** 6
**Confidence:** 3

**Summary:**

In this paper, the authors focus on the lightweight image super-resolution problem. The main contribution of this paper is exploring a flexible meta pruning technique which combines the structured and unstructured network pruning. Experimental results indicate the superiority of the proposed method as compared to existing lightweight image super-resolution methods.

**Strengths:**

1.	The proposed flexible meta pruning technique combines structured and unstructured network pruning, enabling flexible pruning.
2.	The proposed network achieves better performance than most existing lightweight image super-resolution methods,

**Weaknesses:**

1.	The introduction of the proposed methods appears somewhat confusing. The proposed pruning method combines both structured and unstructured pruning, but it is difficult to distinguish their specific implementations and how they are integrated when reading this paper. Are channels being pruned and weights set to zero, or is it only utilizing sparse regularization for structural pruning.
2.	The comparison with existing methods lacks a comparison of runtime.
3.	Table 6 is challenging to understand.
4.	The advantages of this method in terms of flexibility compared to standalone structured and unstructured pruning need to be further emphasized.

==============After rebuttal===========
The authors responses address most of my concerns, so I am willing to increase the score to 6

**Questions:**

1.	The proposed pruning method combines both structured and unstructured pruning, but it is difficult to distinguish their specific implementations and how they are integrated when reading this paper. Are channels being pruned and weights set to zero, or is it only utilizing sparse regularization for structural pruning.
2.	As a pruning method, how does it perform when applied to existing super-resolution networks?

---

> ### Author Response · Authors · 2023-11-20
> **Response to Reviewer pKAh (denoted as R5) part 1**
>
> `Q5-1:` The introduction of the proposed methods appears somewhat confusing. The proposed pruning method combines both structured and unstructured pruning, but it is difficult to distinguish their specific implementations and how they are integrated when reading this paper. Are channels being pruned and weights set to zero, or is it only utilizing sparse regularization for structural pruning.
>
> `A5-1:`  Thank you for your valuable feedback.
>
> **(1)** Specific implementations. We optimize the channel vectors and weight indicators with proximal gradient and SGD. There are mainly two reasons: **First**, SGD with regularization could already lead to acceptable weight sparsity while the channel sparsity could be harder to achieve. Thus, an advanced algorithm proximal gradient descent (PGD) is used for channel pruning to accelerate the convergence of the optimization procedure. **Second**, by separating the optimization, we are more flexible in controlling the contribution of channel pruning and weight pruning.
>
> **(2)** How they are integrated? During the optimization, we set regularization factors $\lambda_C$=0.1 and $\lambda_W$=0.02 for both structured pruning and unstructured pruning. If the compression ratio of one pruning method is achieved, the optimization method for either the channel vectors or the weight indicators is stopped while the other one continues. The whole algorithm converges if both of the compression targets are achieved.
>
> **(3)** Network pruning output. The pruned channels are removed. The pruned weights are set to zero. We conduct both structural and unstructural pruning.
>
> We will revise the paper and make it clearer based on your comments.
>
> `Q5-2:` The comparison with existing methods lacks a comparison of runtime.
>
> `A5-2:` Thanks for the valuable suggestion. We provide inference time with other methods in the following Table. EDSR-16-256 means 16 residual blocks (RBs) with 256 channels for each Conv layer in RB. The inference time is tested on Urban100 ($\times$4) with an RTX 3090.
>
>    |  Method   | EDSR-16-256 |   CARN   |   IMDN   |  DHP   |  ASSLN |  FMP (ours) |
>    |  -------  |  :-------:  | :------: | :------: | :----: | :----: | :----:      |
>    | Time (ms) |     1397    |    53    |     39   |   49   |  49    |   21        |
>
> According to the inference time comparison, we can see that our FMP achieves less inference time than others.
>
> `Q5-3:` Table 6 is challenging to understand.
>
> `A5-3:` Thanks for the valuable comment.
>
>
> **(1)** Table 6 is about the results of different convergence criteria. We provided explanations about convergence criteria in Sec. 4.4 of the main paper.
>
> **(2)** In Sec. 4.4, we provided a brief analysis about Table 6. Meanwhile, in Sec. 2.2 of the supplementary material, we provided further analyses about convergence criteria.
>
> Based on the reviewer's comments, we will add more content from supplementary material to the main paper, trying to make Table 6 easier to understand.
>
>
> `Q5-4:` The advantages of this method in terms of flexibility compared to standalone structured and unstructured pruning need to be further emphasized.
>
> `A5-4:` Thanks for the valuable suggestion.
>
> **(1)** In Sec. 3.1 of the main paper, we give some reasons why we need flexible pruning. This is a part of our motivation, where we also briefly discuss the advantages of our FMP over standalone structured and unstructured pruning.
>
> **(2)** In Sec. 1.1 of the supplementary material, we provide more clarifications and explanations about the motivation of our method.
>
> We will revise Sec. 3.1 of the main paper to further emphasize our advantages as suggested by the reviewer.

---

> ### Author Response · Authors · 2023-11-20
> **Response to Reviewer pKAh (denoted as R5) part 2**
>
> `Q5-5:` The proposed pruning method combines both structured and unstructured pruning, but it is difficult to distinguish their specific implementations and how they are integrated when reading this paper. Are channels being pruned and weights set to zero, or is it only utilizing sparse regularization for structural pruning.
>
> `A5-5:` Thanks for the valuable question.
>
> **(1)** Specific implementations. We optimize the channel vectors and weight indicators with proximal gradient and SGD. There are mainly two reasons: **First**, SGD with regularization could already lead to acceptable weight sparsity while the channel sparsity could be harder to achieve. Thus, an advanced algorithm proximal gradient descent (PGD) is used for channel pruning to accelerate the convergence of the optimization procedure. **Second**, by separating the optimization, we are more flexible in controlling the contribution of channel pruning and weight pruning.
>
> **(2)** How they are integrated? During the optimization, we set regularization factors $\lambda_C$=0.1 and $\lambda_W$=0.02 for both structured pruning and unstructured pruning. If the compression ratio of one pruning method is achieved, the optimization method for either the channel vectors or the weight indicators is stopped while the other one continues. The whole algorithm converges if both of the compression targets are achieved.
>
> **(3)** Network pruning output. The pruned channels are removed. The pruned weights are set to zero. We conduct both structural and unstructural pruning.
>
> We will revise the paper and make it clearer based on your comments.
>
> `Q5-6:` As a pruning method, how does it perform when applied to existing super-resolution networks?
>
> `A5-6:` Thanks for the valuable question about the generalization ability of our method.
>
> **(1)** In this work, we mainly apply our FMP to CNN-based EDSR and newly designed LSRB. To further demonstrate the generalization ability of the network pruning methods, we should also have applied FMP to several classic image SR models. Although LSRB is closely related to several leading lightweight networks (e.g., IMDN, RFAN, RLFN), it is more convincing to try more SR models.
>
> **(2)** It is worth investigating FMP in Transformer-based methods (e.g., SwinIR).
>
> **(3)** Our FMP is a general idea for flexible structured and unsructred pruning. We can apply FMP to CNN-based and Transformer-based image SR methods.
>
> We will further investigate such a direction and provide more baselines.

---

> > ### Comment · Reviewer_pKAh · 2023-11-22
> > **Response to authors**
> >
> > Many thanks to authors for their careful responses. I think the authors provide adequate explanations for most of my questions, so I am willing  to raise my rating score to 6.

---

### Official Review · Reviewer_uZjY · 2023-10-28

**Soundness:** 3 good
**Presentation:** 3 good
**Contribution:** 2 fair
**Rating:** 5
**Confidence:** 5

**Summary:**

The manuscript introduces a novel approach, Flexible Meta Pruning (FMP), designed for lightweight image super-resolution. FMP entails the incorporation of network channel vectors and weight tensors into a hypernetwork, facilitating simultaneous pruning and optimization through proximal gradient and SGD. Additionally, the manuscript substitutes the residual block in RLFN with a simplified version and introduces the Lightweight SR Baseline (LSRB). The application of FMP to LSRB demonstrates competitive performances with relatively low computational complexity in several super-resolution experiments.

**Strengths:**

(1)	The manuscript is presented clearly and well-structured, complemented by easily understandable figures and charts.
(2)	The concept of concurrently combining multiple pruning techniques is intuitive, and the proposed method is straightforward to comprehend and follow.
(3)	The manuscript includes rich ablation studies, and experimental results effectively showcases the superior visual performance achieved by the proposed method.

**Weaknesses:**

(1)	The manuscript's contribution seems to be incremental, primarily involving the addition of kernel weight pruning based on DHP [1]. While the primary distinction between the proposed method and DHP lies in kernel pruning, the results shown in Table 6 indicate that simultaneous channel and kernel pruning may not significantly show effectiveness over channel pruning alone.
(2)	Some of the experimental comparisons may be perceived as lacking in both fairness and completeness when it comes to demonstrating the effectiveness of the method. Notably, in Figure 1, the analysis exclusively focuses on showcasing the influence of FMP. However, it's important to note that the baseline models utilized for evaluating FMP and other comparative methods differ. This dissimilarity in the baseline models pose challenges when attempting to assess the precise impact of FMP. Besides, in subsection 4.2, the experimental comparisons fail to cover some classic lightweight SR models including RLFN and ELAN, which are also mentioned in Section 1 and Section 2
(3)	The manuscript does not sufficiently address the compatibility and generalizability of the proposed FMP method. Although FMP was developed for lightweight super-resolution, experiments have only been conducted on one model, LSRB. The suitability of applying FMP to other super-resolution models remains unexplained.

Reference
[1] Li Y, Gu S, Zhang K, et al. Dhp: Differentiable meta pruning via hypernetworks[C]//Computer Vision–ECCV 2020: 16th European Conference, Glasgow, UK, August 23–28, 2020, Proceedings, Part VIII 16. Springer International Publishing, 2020: 608-624.

**Questions:**

(1)	What optimization method was employed for the FMP method outlined in the manuscript? While the introduction section mentions SGD and proximal gradient descent, the experimental setup section suggests the use of Adam for FMP. This inconsistency is expected to be clarified.
(2)	LSFB represents a lightweight network achieved by replacing residual blocks in RLFN with simplified versions, and FMP is a lightweight method. Table 3 indicates that LSFB+FMP exhibits higher model parameters and computational complexity compared to RLFN. Could you provide an explanation for this discrepancy? Furthermore, LSFB+FMP demonstrates increased inference speed, despite the primary goal of parameter optimization not being fast adaptation in meta-learning. The reasons behind the improved inference speed is expected to be addressed and analyzed.
(3)	FMP introduces kernel weight pruning in addition to DHP, and the results in Table 4 is intended to show that FMP outperformances DHP. However, the combination of multiple pruning techniques may not always lead to an improved result. Besides, it seems that the higher the prune ratio, the better for the performances of both DHP and FMP. Both appear to be counterintuitive to some extents. Therefore, more detailed analysis and explanation are necessary here.
(4)	Formula (6) is similar to formula (7) because it needs to calculate the norm as well. However, why not trying to choose the suitable norm through experiments in the same way adopted to determine the norm in formula (7)?
(5)	In ESA module of LSRB, spatial attention is used instead of convolution. Is this module involved in the proposed pruning method?

---

> ### Author Response · Authors · 2023-11-20
> **Response to Reviewer uZjY (denoted as R4) part 1**
>
> `Q4-1:` The manuscript's contribution seems to be incremental, primarily involving the addition of kernel weight pruning based on DHP [1]. While the primary distinction between the proposed method and DHP lies in kernel pruning, the results shown in Table 6 indicate that simultaneous channel and kernel pruning may not significantly show effectiveness over channel pruning alone.
>
> `A4-1:` Thank you for your valuable feedback.
>
> **(1)** In the first sentence of Sec. 4.3 of main paper, we state that training process will stop if meets convergence or reach the maximum iterations 300K. We find that our FMP would meet convergence with fewer iterations than DHP. But, this is mainly based on the validation performance to early stop the training.
>
> **(2)** When we stop training our FMP with fewer iterations than DHP, our model may not achieve obvious improvements than DHP on other datasets, like Urban100.
>
> **(3)** We allow early stop for the ablation study is mainly to save the training time with limited resource. We hope to investigate if it is possible for us to train the lightweight models with fewer iterations, which we think is also favorable in real practice.
>
> Of course, we will revise the paper and make this part clearer based on the reviewer's comments.
>
>
> `Q4-2:` Some of the experimental comparisons may be perceived as lacking in both fairness and completeness when it comes to demonstrating the effectiveness of the method. Notably, in Figure 1, the analysis exclusively focuses on showcasing the influence of FMP. However, it's important to note that the baseline models utilized for evaluating FMP and other comparative methods differ. This dissimilarity in the baseline models pose challenges when attempting to assess the precise impact of FMP. Besides, in subsection 4.2, the experimental comparisons fail to cover some classic lightweight SR models including RLFN and ELAN, which are also mentioned in Section 1 and Section 2.
>
> `A4-2:` Thanks for the valuable suggestions.
>
> **(1)** In Figure 1, we mainly want to give an overall view of our method FMP. Specifically, we show that an image SR baseline with FMP can achieve comparable or better results than recent methods.
>
> **(2)** To showcase the influence of FMP, we main provide corresponding results in ablation part.
>
> **(3)** We compare with more recent works, like SRPN [Ref1], RLFN [Ref2], and ELAN [Ref3]. We provide PSNR/SSIM values of compared methods in the following table.
>
>    |  Method  |   Scale    |    Set5        |   Set14        |   B100         |  Urban100      |
>    |  ------- | :-------:  | :-------:      | :------:       | :------:       | :----:         |
>    |   SRPN   | $\times$2  | 38.10 / 0.9608 | 33.70 / 0.9189 | 32.25 / 0.9005 | 32.26 / 0.9294 |
>    |   RLFN   | $\times$2  | 38.07 / 0.9607 | 33.72 / 0.9187 | 32.22 / 0.9000 | 32.33 / 0.9299 |
>    |   ELAN   | $\times$2  | 38.17 / 0.9611 | 33.94 / 0.9207 | 32.30 / 0.9012 | 32.76 / 0.9340 |
>    |FMP (ours)| $\times$2  | 38.17 / 0.9615 | 33.81 / 0.9215 | 32.32 / 0.9022 | 32.71 / 0.9360 |
>    |   SRPN   | $\times$3  | 34.47 / 0.9276 | 30.38 / 0.8425 | 29.16 / 0.8061 | 28.22 / 0.8534 |
>    |   RLFN   | $\times$3  | N/A            | N/A            | N/A            | N/A            |
>    |   ELAN   | $\times$3  | 34.61 / 0.9288 | 30.55 / 0.8463 | 29.21 / 0.8081 | 28.69 / 0.8624 |
>    |FMP (ours)| $\times$3  | 34.55 / 0.9291 | 30.48 / 0.8456 | 29.20 / 0.8101 | 28.40 / 0.8597 |
>    |   SRPN   | $\times$4  | 32.24 / 0.8958 | 28.69 / 0.7836 | 27.63 / 0.7373 | 26.16 / 0.7875 |
>    |   RLFN   | $\times$4  | 32.24 / 0.8952 | 28.62 / 0.7813 | 27.60 / 0.7364 | 26.17 / 0.7877 |
>    |   ELAN   | $\times$4  | 32.43 / 0.8975 | 28.78 / 0.7858 | 27.69 / 0.7406 | 26.54 / 0.7982 |
>    |FMP (ours)| $\times$4  | 32.34 / 0.8979 | 28.71 / 0.7878 | 27.67 / 0.7425 | 26.35 / 0.7954 |
>
> According to the table, we can see that our FMP achieves comparable performance than recent leading methods.
>
> We will revise the paper with more recent methods based on the suggestios.
>
> [Ref1] Learning Efficient Image Super-Resolution Networks via Structure-Regularized Pruning, ICLR, 2022
>
> [Ref2] Residual Local Feature Network for Efficient Super-Resolution, CVPRW, 2022
>
> [Ref3] Efficient Long-range Attention Network for Image Super-Resolution, ECCV, 2022

---

> ### Author Response · Authors · 2023-11-20
> **Response to Reviewer uZjY (denoted as R4) part 2**
>
> `Q4-3:` The manuscript does not sufficiently address the compatibility and generalizability of the proposed FMP method. Although FMP was developed for lightweight super-resolution, experiments have only been conducted on one model, LSRB. The suitability of applying FMP to other super-resolution models remains unexplained.
>
> `A4-3:` Thanks for the valuable suggestion.
>
> **(1)** In this work, we mainly apply our FMP to CNN-based EDSR and newly designed LSRB. To further demonstrate the generalization ability of the network pruning methods, we should also have applied FMP to several classic image SR models. Although LSRB is closely related to several leading lightweight networks (e.g., IMDN, RFAN, RLFN), it is more convincing to try more SR models. We can mainly replace the LSRB with other SR methods and use hypernetwork to generate their corresponding parameters.
>
> **(2)** It is worth investigating FMP in Transformer-based methods (e.g., SwinIR).
>
> **(3)** Our FMP is a general idea for flexible structured and unsructred pruning. We can apply FMP to CNN-based and Transformer-based image SR methods.
>
> We will further investigate such a direction and provide more baselines.
>
> `Q4-4:` What optimization method was employed for the FMP method outlined in the manuscript? While the introduction section mentions SGD and proximal gradient descent, the experimental setup section suggests the use of Adam for FMP. This inconsistency is expected to be clarified.
>
> `A4-4:` Thanks for pointing this part out.
>
> **(1)** We optimize the channel vectors and weight indicators with proximal gradient and SGD. There are mainly two reasons: **First**, SGD with regularization could already lead to acceptable weight sparsity while the channel sparsity could be harder to achieve. Thus, an advanced algorithm proximal gradient descent (PGD) is used for channel pruning to accelerate the convergence of the optimization procedure. **Second**, by separating the optimization, we are more flexible in controlling the contribution of channel pruning and weight pruning.
>
> **(2)** When we train the image SR network with flexible meta pruning jointly, we have to update the **whole network parameters**. In this procesure, we use the general ADAM optimizer, which is also adopted in most of other image SR methods.
>
> We will further clarify them in the paper based on the suggestion.
>
> `Q4-5:` LSFB represents a lightweight network achieved by replacing residual blocks in RLFN with simplified versions, and FMP is a lightweight method. Table 3 indicates that LSFB+FMP exhibits higher model parameters and computational complexity compared to RLFN. Could you provide an explanation for this discrepancy? Furthermore, LSFB+FMP demonstrates increased inference speed, despite the primary goal of parameter optimization not being fast adaptation in meta-learning. The reasons behind the improved inference speed is expected to be addressed and analyzed.
>
> `A4-5:` Thanks for the valuable suggestion.
>
> In Table 3 of the main paper, our LSFB has slightly larger number of parameters and FLOPs, but achieves faster speed. The reasons are mainly summarized as follows:
>
> **(1)** In the build block RLRB in RLFN, there are three convolutional layers (each layer is followed by ReLU) with channel number set as 52. While, the building block in our LSFB-6-48 has two convoluional layers (only the first convolutional layer is followed by ReLU) with channel number as 48.
>
> **(2)** Currently, in PyTorch, as used in our work, networks with wider width (i.e., larger channel number) will run slower than narrower one with similar model size. So, LSFB runs faster than RLFN.
>
> **(3)** As we state in Effectiveness of LSFB in Sec. 4.3, it is promising to further reduce its redundant parameters with our proposed FMP. In our research, we can hardly use our common GPU with PyTorch to achieve further accelaration. But, the inference time can be further improved by AI accelerators, which are specifically designed for unstructured pruning.
>
> We will include those clarifications and explanations in the revised paper.

---

> ### Author Response · Authors · 2023-11-20
> **Response to Reviewer uZjY (denoted as R4) part 3**
>
> `Q4-5:` LSRB represents a lightweight network achieved by replacing residual blocks in RLFN with simplified versions, and FMP is a lightweight method. Table 3 indicates that LSRB+FMP exhibits higher model parameters and computational complexity compared to RLFN. Could you provide an explanation for this discrepancy? Furthermore, LSRB+FMP demonstrates increased inference speed, despite the primary goal of parameter optimization not being fast adaptation in meta-learning. The reasons behind the improved inference speed is expected to be addressed and analyzed.
>
> `A4-5:` Thanks for the valuable suggestion.
>
> In Table 3 of the main paper, our LSRB has slightly larger number of parameters and FLOPs, but achieves faster speed. The reasons are mainly summarized as follows:
>
> **(1)** In the building block RLRB in RLFN, there are three convolutional layers (each layer is followed by ReLU) with channel number set as 52. While, the building block in our LSRB-6-48 has two convolutional layers (only the first convolutional layer is followed by ReLU) with channel number as 48.
>
> **(2)** Currently, in PyTorch, as used in our work, networks with wider width (i.e., larger channel number) will run slower than narrower one with similar model size. So, LSRB runs faster than RLFN.
>
> **(3)** As we state in Effectiveness of LSRB in Sec. 4.3, it is promising to further reduce its redundant parameters with our proposed FMP. In our research, we can hardly use our common GPU with PyTorch to achieve further acceleration. But, the inference time can be further improved by AI accelerators, which are specifically designed for unstructured pruning.
>
> We will include those clarifications and explanations in the revised paper.
>
> `Q4-6:` FMP introduces kernel weight pruning in addition to DHP, and the results in Table 4 is intended to show that FMP outperforms DHP. However, the combination of multiple pruning techniques may not always lead to an improved result. Besides, it seems that the higher the prune ratio, the better for the performances of both DHP and FMP. Both appear to be counterintuitive to some extents. Therefore, more detailed analysis and explanation are necessary here.
>
> `A4-6:` Thanks for the valuable comments and suggestions.
>
> **(1)** In the first sentence of Sec. 4.3 of main paper, we state that the training process will stop if meets convergence or reach the maximum iterations 300K. We find that our FMP would meet convergence with fewer iterations than DHP. But, this is mainly based on the validation performance to early stop the training.
>
> **(2)** When we stop training our FMP with fewer iterations than DHP, our model may not achieve obvious improvements than DHP on other datasets, like Urban100.
>
> **(3)** We allow early stop for the ablation study is mainly to save the training time with limited resource. We hope to investigate if it is possible for us to train the lightweight models with fewer iterations, which we think is also favorable in real practice.
>
> **(4)** When the prune ratio becomes higher, the training needs more iterations. Models with more training iterations would actually get higher performance.
>
> We will revise the paper with more detailed analyses and explanations.
>
> `Q4-7:` Formula (6) is similar to formula (7) because it needs to calculate the norm as well. However, why not trying to choose the suitable norm through experiments in the same way adopted to determine the norm in formula (7)?
>
> `A4-7:` Thanks for the interesting question. The main reasons are as follows:
>
> **(1)** Formula (6) is for channel pruning, which has been widely investigated before in image SR, like DHP and ASSL. In previous work, we find that $L_1$ norm is a good choice for channel pruning.
>
> **(2)** We choose $L_1$ norm for channel pruning so that we can keep the same as previous channel pruning works.
>
> **(3)** Unstructured pruning is seldom investigated in image SR before. So, we try different choices of norm and determine the final one for our experiments.
>
> So, in our work, we mainly investigate the choices of norm for Eq. (7), namely unstructured pruning.
>
> `Q4-8:` In ESA module of LSRB, spatial attention is used instead of convolution. Is this module involved in the proposed pruning method?
>
> `A4-8:` Thanks for the question. Yes, this module is involved in our pruning method.

---

> > ### Comment · Reviewer_uZjY · 2023-11-22
> >
> > I would like to thank the authors for providing the response. However, after a thorough reading and check on the response, I am still concerned, especially about the extent of technical contribution and the generalization by just adopting FMP into LSRB, which has still not been clearly shown and proven by the experimental comparisons with SOTA methods. Therefore I would like to keep my score.

---

### Official Review · Reviewer_6gG5 · 2023-10-30

**Soundness:** 3 good
**Presentation:** 4 excellent
**Contribution:** 4 excellent
**Rating:** 8
**Confidence:** 4

**Summary:**

This paper proposes a flexible meta pruning (FMP) for lightweight image super-resolution (SR). The basic idea of FMP is interesting and consists of structured and unstructured pruning simultaneously during the SR network training. The authors propose channel vectors and weight indicators to control channel and weight sparsity of SR network. A simple yet effective SR backbone (LSRB) is also designed. Extensive ablations show the effect of several key structures and methods, like LSRB, flexible pruning, pruning method. The main comparisons with others also show that the proposed FMP achieves good performance.

**Strengths:**

The idea about flexible meta pruning (FMP) is straightforward but effective. FMP conduct structured and unstructured pruning simultaneously in the SR training, which makes the pruning and SR reconstruction more optimized.

The authors propose channel vectors and weight indicators to control network sparsity and optimize them with proximal gradient and SGD respectively. Such different optimization methods make the differentiable and flexible pruning easily.

Extensive ablation studies are conducted to show the effect of each proposed key component, like effect of the baseline LSRB, pruning method, flexible vs. channel pruning.

The authors compare with recent related methods and achieve better performance with both quantitative and visual results. The authors provide PSNR/SSIM, visual results, and also model complexity analyses. Those results further support the claimed contributions.

The authors further discuss convergence criteria and compare with other model compression methods (e.g., NAS, KD, and channel pruning). In this comparison, the proposed FMP still show obvious gains over others.

The paper is well-written and easy to follow. The overall paper is well-organized.

The authors promise to release code, which makes the reproducibility easier and the experiments more solid.

**Weaknesses:**

The performance gains shown in Table 4 seem to be marginal. Did the results of DHP and FMP obtained by using same number of iterations or not? According to the paper, it seems that they may use different iterations based on the convergence criteria. If FMP uses less iterations and achieves similar or better results than DHP, then it makes sense. Please clarify this.

The authors mainly apply FMP to CNN based networks, like EDSR and LSRB. It is also important to apply this method to other structures, like Transformer-based methods. So that, we can see how well its generalization ability is.

When applying the proposed FMP to other SR methods, do people have to modify too much parts of the new models? It seems that FMP can also be used for other image restoration methods. If so, it is better to discuss in the paper.

More recent methods can be compared. For example, ICLR-2022-SRPN can also be discussed and compared.
[ICLR-2022-SRPN] Zhang et al., Learning Efficient Image Super-Resolution Networks via Structure-Regularized Pruning, ICLR, 2022

**Questions:**

The performance is obviously better than others. Did the authors used self-ensemble during the test phase?

In practice, does FMP save more running time than channel or weight pruning? Or does FMP need specific hardware to reach obvious reduction of resource (e.g., GPU memory and running time)?

---

> ### Author Response · Authors · 2023-11-20
> **Response to Reviewer 6gG5 (denoted as R3) part 1**
>
> `Q3-1:` The performance gains shown in Table 4 seem to be marginal. Did the results of DHP and FMP obtained by using same number of iterations or not? According to the paper, it seems that they may use different iterations based on the convergence criteria. If FMP uses less iterations and achieves similar or better results than DHP, then it makes sense. Please clarify this.
>
> `A3-1:` Thanks for the valuable comments.
>
> **(1)** In the first sentence of Sec. 4.3 of the main paper, we state that the training process will stop if meets convergence or reaches the maximum iterations 300K. We find that our FMP would meet convergence with fewer iterations than DHP. But, this is mainly based on the validation performance to early stop the training.
>
> **(2)** When we stop training our FMP with fewer iterations than DHP, our model may not achieve obvious improvements than DHP on other datasets, like Urban100.
>
> **(3)** We allow an early stop for the ablation study mainly to save training time with limited resources. We hope to investigate if it is possible for us to train lightweight models with fewer iterations, which we think is also favorable in real practice.
>
> **(4)** When the prune ratio becomes higher, the training needs more iterations. Models with more training iterations would actually get higher performance.
>
> We will clarify them in the revised paper with more detailed analyses and explanations.
>
>
> `Q3-2:` The authors mainly apply FMP to CNN based networks, like EDSR and LSRB. It is also important to apply this method to other structures, like Transformer-based methods. So that, we can see how well its generalization ability is.
>
> `A3-2:` Thanks for the valuable question about the generalization ability of our method.
>
> **(1)** In this work, we mainly apply our FMP to CNN-based EDSR and newly designed LSRB. To further demonstrate the generalization ability of the network pruning methods, we should also have applied FMP to several classic image SR models. Although LSRB is closely related to several leading lightweight networks (e.g., IMDN, RFAN, RLFN), it is more convincing to try more SR models.
>
> **(2)** It is worth investigating FMP in Transformer-based methods (e.g., SwinIR).
>
> **(3)** Our FMP is a general idea for flexible structured and unstructured pruning. We can apply FMP to CNN-based and Transformer-based image SR methods.
>
> We will further investigate such a direction and provide more baselines.
>
> `Q3-3:` When applying the proposed FMP to other SR methods, do people have to modify too much parts of the new models? It seems that FMP can also be used for other image restoration methods. If so, it is better to discuss in the paper.
>
> `A3-3:` Thanks for the interesting question.
>
> **(1)** In this work, we mainly use image SR as an example to incorporate flexible meta pruning.
>
> **(2)** Our pruning method is a general one and can be easily applied to image restoration methods. For other image restoration models, we can still use FMP to conduct both structured and unstructured pruning. We do not have to make too many modifications.
>
> We will add those discussions to the revised paper.

---

> ### Author Response · Authors · 2023-11-20
> **Response to Reviewer 6gG5 (denoted as R3) part 2**
>
> `Q3-4:` More recent methods can be compared. For example, ICLR-2022-SRPN can also be discussed and compared. [ICLR-2022-SRPN] Zhang et al., Learning Efficient Image Super-Resolution Networks via Structure-Regularized Pruning, ICLR, 2022
>
>
> `A3-4:` Thanks for the valuable suggestions. We compare with more recent works, like SRPN [Ref1], RLFN [Ref2], and ELAN [Ref3]. We provide PSNR/SSIM values of compared methods in the following table.
>
>    |  Method  |   Scale    |    Set5        |   Set14        |   B100         |  Urban100      |
>    |  ------- | :-------:  | :-------:      | :------:       | :------:       | :----:         |
>    |   SRPN   | $\times$2  | 38.10 / 0.9608 | 33.70 / 0.9189 | 32.25 / 0.9005 | 32.26 / 0.9294 |
>    |   RLFN   | $\times$2  | 38.07 / 0.9607 | 33.72 / 0.9187 | 32.22 / 0.9000 | 32.33 / 0.9299 |
>    |   ELAN   | $\times$2  | 38.17 / 0.9611 | 33.94 / 0.9207 | 32.30 / 0.9012 | 32.76 / 0.9340 |
>    |FMP (ours)| $\times$2  | 38.17 / 0.9615 | 33.81 / 0.9215 | 32.32 / 0.9022 | 32.71 / 0.9360 |
>    |   SRPN   | $\times$3  | 34.47 / 0.9276 | 30.38 / 0.8425 | 29.16 / 0.8061 | 28.22 / 0.8534 |
>    |   RLFN   | $\times$3  | N/A            | N/A            | N/A            | N/A            |
>    |   ELAN   | $\times$3  | 34.61 / 0.9288 | 30.55 / 0.8463 | 29.21 / 0.8081 | 28.69 / 0.8624 |
>    |FMP (ours)| $\times$3  | 34.55 / 0.9291 | 30.48 / 0.8456 | 29.20 / 0.8101 | 28.40 / 0.8597 |
>    |   SRPN   | $\times$4  | 32.24 / 0.8958 | 28.69 / 0.7836 | 27.63 / 0.7373 | 26.16 / 0.7875 |
>    |   RLFN   | $\times$4  | 32.24 / 0.8952 | 28.62 / 0.7813 | 27.60 / 0.7364 | 26.17 / 0.7877 |
>    |   ELAN   | $\times$4  | 32.43 / 0.8975 | 28.78 / 0.7858 | 27.69 / 0.7406 | 26.54 / 0.7982 |
>    |FMP (ours)| $\times$4  | 32.34 / 0.8979 | 28.71 / 0.7878 | 27.67 / 0.7425 | 26.35 / 0.7954 |
>
> According to the table, we can see that our FMP achieves comparable performance to recent leading methods.
>
> We will revise the paper with more recent methods based on the suggestions.
>
> [Ref1] Learning Efficient Image Super-Resolution Networks via Structure-Regularized Pruning, ICLR, 2022
>
> [Ref2] Residual Local Feature Network for Efficient Super-Resolution, CVPRW, 2022
>
> [Ref3] Efficient Long-range Attention Network for Image Super-Resolution, ECCV, 2022
>
> `Q3-5:` The performance is obviously better than others. Did the authors use self-ensemble during the test phase?
>
> `A3-5:` Thanks for pointing this out. No, we did not use self-ensemble during the test phase. In Sec. 4.2 of the main paper, we have stated that we provide our quantitative results without self-ensemble.
>
> `Q3-6:` In practice, does FMP save more running time than channel or weight pruning? Or does FMP need specific hardware to reach obvious reduction of resource (e.g., GPU memory and running time)?
>
> `A3-6:` Thanks for the valuable comments.
>
> **(1)** Currently, with PyTorch and a common GPU (e.g., RTX 3090), our FMP can hardly save more running time than channel pruning. This is mainly because unstructured pruning is not specifically optimized and can hardly achieve further acceleration.
>
> **(2)** But, the inference time can be further improved by AI accelerators, which are specifically designed for unstructured pruning.
>
> We will include those discussions in the revised paper.

---

> > ### Comment · Reviewer_6gG5 · 2023-11-22
> > **Feedback**
> >
> > Thanks for your detailed responses. My concerns have been well solved. Thus, I tend to keep my original score as "8: accept"

---

### Official Review · Reviewer_znMB · 2023-11-01

**Soundness:** 3 good
**Presentation:** 3 good
**Contribution:** 3 good
**Rating:** 6
**Confidence:** 4

**Summary:**

This paper proposes a flexible mera pruning method named FMP, which combines the merits of both structural and unstructured pruning, leading to better trade-off between accuracy and latency.

**Strengths:**

1. The idea of combining the merits of structured and unstructured pruning is straightforward and promising.

2. The experiments show that the proposed method can surpass the competitors with fewer parameters and less latency.

3. The paper is well written and the idea is clearly illustrated.

**Weaknesses:**

1. The authors argue that unstructured pruning does not contribute to actual acceleration. However, the experiments in Table 1 do not include comparisons of latency.

2. Some leading lightweight methods are missed in Table 1, e.g. MegSR[1], VapSR[2], Omni-SR[3]. The authors are suggested to provide comparisons over varied model sizes to verify the effectiveness of the proposed method, because we can not justify whether it can maintain its advantage among smaller models (~400 K params).

3. Table 4 compares only one pruning method because other methods use pretrained SR backbones. However, to better justify the effectiveness of the proposed method, the authors can add more comparisons by also using a pretrained backbone. The pretrained backbone does not increase the inference cost and should not be viewed as a drawback of lightweight SR methods.

4. According to Table 3, the backbone used by FMP already brings some improvements. Thus comparisons in Table 7 may be not fair. The authors are suggested to provide the results of applying ASSLN to  LSRB.

[1] Yu et al., DIPNet: Efficiency Distillation and Iterative Pruning for Image Super-Resolution, 2023

[2] zhou et al., Efficient Image Super-Resolution using Vast-Receptive-Field Attention, 2022

[3] wang et al.,  Omni Aggregation Networks for Lightweight Image Super-Resolution, 2023

**Questions:**

See the weakness above.

---

> ### Author Response · Authors · 2023-11-21
> **Response to Reviewer znMB (denoted as R2) part 1**
>
> `Q2-1:` The authors argue that unstructured pruning does not contribute to actual acceleration. However, the experiments in Table 1 do not include comparisons of latency.
>
> `A2-1:` Thank you for your valuable feedback.
>
> **(1)** In Table 1 of the main paper, it is not easy to provide inference time for all the compared methods. Many of them are officially implemented with different frameworks. For example, SRCNN, FSRCNN, VDSR, DRCN, and MemNet are officially implemented with Caffe. LapSRN is officially implemented with MatConvNet. To compare their inference time fairly, we should implement them under the same codebase and framework and run the models with the same GPU. Currently, we do not have such a codebase. But, it is important to have such a benchmark codebase in the future.
>
> **(2)** Instead, we provide inference time with other methods in the following Table. EDSR-16-256 means 16 residual blocks (RBs) with 256 channels for each Conv layer in RB. The inference time is tested on Urban100 ($\times$4) with an RTX 3090.
>
>    |  Method   | EDSR-16-256 |   CARN   |   IMDN   |  DHP   |  ASSLN |  FMP (ours) |
>    |  -------  |  :-------:  | :------: | :------: | :----: | :----: | :----:      |
>    | Time (ms) |     1397    |    53    |     39   |   49   |  49    |   21        |
>
> According to the inference time comparison, we can see that our FMP achieves less inference time than others.
>
> `Q2-2:` Some leading lightweight methods are missed in Table 1, e.g. DIPNet[1], VapSR[2], Omni-SR[3]. The authors are suggested to provide comparisons over varied model sizes to verify the effectiveness of the proposed method, because we can not justify whether it can maintain its advantage among smaller models (~400 K params).
>
> [1] Yu et al., DIPNet: Efficiency Distillation and Iterative Pruning for Image Super-Resolution, CVPR Workshop, 2023
>
> [2] zhou et al., Efficient Image Super-Resolution using Vast-Receptive-Field Attention, ECCV Workshop, 2022
>
> [3] wang et al., Omni Aggregation Networks for Lightweight Image Super-Resolution, CVPR, 2023
>
> `A2-2:` Thanks for the valuable suggestions.
>
> We compare with DIPNet[1], VapSR[2], and Omni-SR[3]. PSNR/SSIM values are provided in the following table. It should be noted that we cite the results of DIPNet, VapSR, and Omni-SR from their papers.
>
>    |  Method             |   Scale    | Params (K)|   Set5  |   Set14        |   B100         |  Urban100      |
>    |  -------            | :-------:  |  :-------:|:-------:| :------:       | :------:       | :----:         |
>    |  DIPNet (CVPRW'23)  | $\times$2  | 527 |37.98 / 0.9605 | 33.66 / 0.9192 | 32.20 / 0.9002 | 32.31 / 0.9302 |
>    |   VapSR (ECCVW'22)  | $\times$2  | 329 |38.08 / 0.9612 | 33.77 / 0.9195 | 32.27 / 0.9011 | 32.45 / 0.9316 |
>    | Omni-SR (CVPR'23)   | $\times$2  | 772 |38.29 / 0.9617 | 34.27 / 0.9238 | 32.41 / 0.9026 | 33.30 / 0.9386 |
>    |   FMP (ours)        | $\times$2  | 694 |38.17 / 0.9615 | 33.81 / 0.9215 | 32.32 / 0.9022 | 32.71 / 0.9360 |
>    |  DIPNet (CVPRW'23)  | $\times$3  | N/A |N/A            | N/A            | N/A            | N/A            |
>    |   VapSR (ECCVW'22)  | $\times$3  | 337 |34.52 / 0.9284 | 30.53 / 0.8452 | 29.19 / 0.8077 | 28.43 / 0.8583 |
>    | Omni-SR (CVPR'23)   | $\times$3  | 780 |34.77 / 0.9304 | 30.70 / 0.8489 | 29.33 / 0.8111 | 29.12 / 0.8712 |
>    |   FMP (ours)        | $\times$3  | 684 |34.55 / 0.9291 | 30.48 / 0.8456 | 29.20 / 0.8101 | 28.40 / 0.8597 |
>    |  DIPNet (CVPRW'23)  | $\times$4  | 543 |32.20 / 0.8950 | 28.58 / 0.7811 | 27.59 / 0.7364 | 26.16 / 0.7879 |
>    |   VapSR (ECCVW'22)  | $\times$4  | 342 |32.38 / 0.8966 | 28.77 / 0.7852 | 27.68 / 0.7398 | 26.35 / 0.7941 |
>    | Omni-SR (CVPR'23)   | $\times$4  | 792 |32.57 / 0.8993 | 28.95 / 0.7898 | 27.81 / 0.7439 | 26.95 / 0.8105 |
>    |  FMP (ours)         | $\times$4  | 704 |32.34 / 0.8979 | 28.71 / 0.7878 | 27.67 / 0.7425 | 26.35 / 0.7954 |
>
> According to the above table, we have the following analyses.
>
> **(1)** All the compared methods (i.e., DIPNet, VapSR, Omni-SR) have new network designs. Those new designs can achieve obviously better performance than our ResNet based backbone LSRB.
>
> **(2)** Network parameter number matters very much. Ommni-SR achieves the best performance with the largest model size.
>
> **(3)** In our work, we mainly focus on baselines with high speed, where we design new models based on the champion solution RLFN at NTIRE'22 Efficient SR. Our FMP can also be applied to new SOTA lightweight image SR methods shown above. It is important and promising to investigate applying our FMP to the above methods in the future.
>
> As suggested by the reviewer, we will further investigate smaller models by pruning more channels and weights.

---

> ### Author Response · Authors · 2023-11-21
> **Response to Reviewer znMB (denoted as R2) part 2**
>
> `Q2-3:` Table 4 compares only one pruning method because other methods use pretrained SR backbones. However, to better justify the effectiveness of the proposed method, the authors can add more comparisons by also using a pretrained backbone. The pretrained backbone does not increase the inference cost and should not be viewed as a drawback of lightweight SR methods.
>
> `A2-3:` Thanks for the valuable suggestions. We adopt ASSLN for comparison. ASSLN needs a pretrained model. We still use EDSR-8-128 as the base model and train ASSLN with the same setting as others. We provide the PSNR values in the following table.
>
>
>    |Prune Ratio|Method|Set5 | Set14  |   B100  |Urban100 |
>    |  -------  | :-------:|:-------:|:------:| :------:| :----:  |
>    |     60    |  ASSLN   |  32.01  | 28.54  | 27.52   | 25.93   |
>    |     60    |   DHP    |  31.99  | 28.52  | 27.53   | 25.92   |
>    |     60    |FMP (ours)|  32.16  | 28.60  | 27.55   | 25.96   |
>    |     40    |  ASSLN   |  32.02  | 28.51  | 27.51   | 25.88   |
>    |     40    |   DHP    |  32.01  | 28.49  | 27.52   | 25.86   |
>    |     40    |FMP (ours)|  32.08  | 28.58  | 27.53   | 25.91   |
>    |     20    |  ASSLN   |  31.95  | 28.45  | 27.48   | 25.71   |
>    |     20    |   DHP    |  31.94  | 28.42  | 27.47   | 25.69   |
>    |     20    |FMP (ours)|  31.97  | 28.51  | 27.47   | 25.78   |
>
> According to the table, we find that ASSLN performs comparable to or slightly better than DHP. Pretrained models would help improve the performance, especially in the current case. Specifically, the training process will stop if meets convergence or reaches the maximum iteration 300K.
>
> `Q2-4:` According to Table 3, the backbone used by FMP already brings some improvements. Thus comparisons in Table 7 may be not fair. The authors are suggested to provide the results of applying ASSLN to LSRB.
>
> `A2-4:` Thanks for the valuable suggestions.
>
> We use the LSRB as the backbone and apply ASSLN to it. As time and GPU resource is limited, we only train the SR model with scale $\times$2. We provide PSNR/SSIM values in the following table.
>
>    |  Method  |   Scale    |    Set5        |   Set14        |   B100         |  Urban100      |
>    |  ------- | :-------:  | :-------:      | :------:       | :------:       | :----:         |
>    |   ASSLN  | $\times$2  | 38.12 / 0.9608 | 33.77 / 0.9194 | 32.27 / 0.9007 | 32.41 / 0.9309 |
>    |ASSLN+LSRB| $\times$2  | 38.13 / 0.9611 | 33.79 / 0.9198 | 32.28 / 0.9010 | 32.52 / 0.9320 |
>    |    FMP   | $\times$2  | 38.17 / 0.9615 | 33.81 / 0.9215 | 32.32 / 0.9022 | 32.71 / 0.9360 |
>
> According to the table, we find that ASSLN+LSRB achieves improvement over the original ASSLN. This observation shows the effectiveness of LSRB. Our FMP still obtains better performance than ASSLN+LSRB. It indicates that flexible pruning (i.e., structured pruning and unstructured pruning) could better reduce network redundancy than channel pruning in image SR.
>
> It should also be noted that ASSLN has to use a large pretrained LSRB model. In contrast, our FMP does not need a pretrained model.

---

> ### Comment · Area_Chair_SFXL · 2023-12-04
> **[Important] Response Required to Authors' Rebuttal**
>
> Dear Reviewer znMB,
>
> As we progress through the review process for ICLR 2024, I would like to remind you of the importance of the rebuttal phase. The authors have submitted their rebuttals, and it is now imperative for you to engage in this critical aspect of the review process.
>
> Please ensure that you read the authors' responses carefully and provide a thoughtful and constructive follow-up. Your feedback is not only essential for the decision-making process but also invaluable for the authors.
>
> Thank you,
>
> ICLR 2024 Area Chair

---

### Official Review · Reviewer_Jqxo · 2023-11-03

**Soundness:** 3 good
**Presentation:** 2 fair
**Contribution:** 3 good
**Rating:** 6
**Confidence:** 3

**Summary:**

This paper proposes the flexible meta pruning (FMP) for lightweight image SR, using a hypernetwork to perform channel pruning and weight pruning simultaneously.
The sparsity of the channels and weights controlled through the channel vectors and weight indicators
Channel vectors and weight indicators are optimized with proximal gradient and SGD.
FMP shows competitive experimental results against leading architectures.

**Strengths:**

This paper combines the channel and weight pruning for model compression in the lightweight image SR and achieves competitive results across multiple datasets when compared to most leading approaches.

**Weaknesses:**

1. It's unclear how the compressed model would translate to the practical inference speed up, considering there are unstructured weight pruning.
2. The ideas of channel and weight pruning have been around for a while. It is not clear what's new with the proposed approach?

**Questions:**

1. What's the benefits of combining channeling pruning with unstructured weight pruning?
2. In Table 3, instead of just the submodule, is it possible to show the inference speed up of the entire model?
3. What's the motivation and benefits of applying proximal gradient for channels vectors and weight indicators?
4. How much performance gain can FMP harvest if using self-ensemble?

---

> ### Author Response · Authors · 2023-11-21
> **Response to Reviewer Jqxo (denoted as R1)**
>
> `Q1-1:` It's unclear how the compressed model would translate to the practical inference speed up, considering there are unstructured weight pruning.
>
> `A1-1:` Thank you for your valuable feedback.
>
> **(1)** Our designed lightweight image SR baseline LSRB reduces inference time than the champion solution RLFN at NTIRE'22 Efficient SR challenge. As shown in Table 3 of the main paper, our LSRB achieves much faster than RLFN.
>
> **(2)** In our work, we can hardly use our common GPU with PyTorch to achieve further acceleration for unstructured pruning. But, the inference time can be further improved by AI accelerators, which are specifically designed for unstructured pruning.
>
> **(3)** We apply FMP to LSRB. With PyTorh and common GPU, the speed up of FMP mainly comes from LSRB and channel pruning. As a result, our FMP can still achieve relative speed up.
>
>
> `Q1-2:` The ideas of channel and weight pruning have been around for a while. It is not clear what's new with the proposed approach?
>
> `A1-2:` Thanks for pointing this out.
>
>  **(1)** Although the idea of combining unstructured pruning and structured pruning for image SR is straightforward, how to design an algorithm to achieve flexible pruning is still worth investigating. Our FMP could automatically allocate parameters and computation budget for unstructured and structured pruning.
>
>  **(2)** The introduced technique such as weight indicator extends the usage of hypernetworks from channel pruning to a wider scope of network (weight and channel) pruning.
>
> `Q1-3:` What's the benefits of combining channeling pruning with unstructured weight pruning?
>
> `A1-3:` Thanks for the question.
>
> **(1)** Structured pruning and unstructured pruning are two important network compression methods that can cut down the model complexity of deep neural networks significantly.
>
> **(2)** As we state in Effectiveness of LSRB in Sec. 4.3, it is promising to further reduce its redundant parameters with our proposed FMP. In our current research, we can hardly use our common GPU with PyTorch to achieve further acceleration.
>
> **(3)** But, the inference time can be further improved by AI accelerators, which are specifically designed for unstructured pruning. As far as we know, there are some specific hardware designs and implementations for unstructured pruning in some institutions, where the inference time can further be reduced. However, we have no access to such hardwares right now.
>
> **(4)** Instead, we provide the potential of further reducing the network redundancy with both structured and unstructured pruning.
>
> `Q1-4:` In Table 3, instead of just the submodule, is it possible to show the inference speed up of the entire model?
>
> `A1-4:` Thanks for the valuable suggestion.
>
> We further provide inference time with other methods in the following Table. EDSR-16-256 means 16 residual blocks (RBs) with 256 channels for each Conv layer in RB. The inference time is tested on Urban100 ($\times$4) with an RTX 3090.
>
>    |  Method   | EDSR-16-256 |   CARN   |   IMDN   |  DHP   |  ASSLN |  FMP (ours) |
>    |  -------  |  :-------:  | :------: | :------: | :----: | :----: | :----:      |
>    | Time (ms) |     1397    |    53    |     39   |   49   |  49    |   21        |
>
> According to the inference time comparison, we can see that our FMP achieves less inference time than others.
>
> `Q1-5:` What's the motivation and benefits of applying proximal gradient for channels vectors and weight indicators?
>
> `A1-5:` Thanks for the interesting question.
>
> **(1)** The general idea of this work is first introduced before elaborating on details of our flexible network pruning method. Structured pruning and unstructured pruning are two important network compression methods that can cut down the model complexity of deep neural networks significantly.
>
> **(2)** Structured pruning leaves regular kernels after pruning, which is beneficial for the actual acceleration of the network.
>
> **(3)** Unstructured pruning removes single weights in a kernel and can compress the network without sacrificing too much accuracy of the network.
>
> **(4)** Since the channel vectors and the weight indicator are not intertwined by non-linear operations, we can decouple their optimization for network pruning by different methods.
>
>
> `Q1-6:` How much performance gain can FMP harvest if using self-ensemble?
>
> `A1-6:` Thanks for pointing this out. Self-ensemle in the test phase is to rotate and flip input, get different output, and then average them to get the final result.
>
> **(1)** Self-ensemble can further achieve about 0.1-0.2 dB in PSNR for different scales and test dataset.
>
> **(2)** However, self-ensemble would use several times of computation cost. For example, in image SR, one common practice of self-ensemble takes 8 times computational cost as the norm one without self-ensemble. So, in our paper, we do not use this strategy for further performance improvement.

---

> ### Comment · Area_Chair_SFXL · 2023-12-04
> **[Important] Response Required to Authors' Rebuttal**
>
> Dear Reviewer Jqxo,
>
> As we progress through the review process for ICLR 2024, I would like to remind you of the importance of the rebuttal phase. The authors have submitted their rebuttals, and it is now imperative for you to engage in this critical aspect of the review process.
>
> Please ensure that you read the authors' responses carefully and provide a thoughtful and constructive follow-up. Your feedback is not only essential for the decision-making process but also invaluable for the authors.
>
> Thank you,
>
> ICLR 2024 Area Chair

---

### Meta-Review · Program_Chairs · 2023-12-06

**Metareview:**

The paper presents an approach in lightweight image super-resolution (SR) by combining channel and weight pruning, and achieves competitive results across multiple datasets. The strengths of the paper lie in its integration of structured and unstructured pruning techniques and its effectiveness demonstrated through extensive experiments. The authors have responded comprehensively to the reviewers' comments, and have addressed many of the concerns raised. However, the paper exhibits some weaknesses, mainly the lack of clarity on how the model's compression translates to practical inference speed-up, particularly due to the unstructured nature of weight pruning. The novelty of the proposed approach, while interesting in its integration of channel and weight pruning, might not be distinctly differentiated from existing methods, and the paper requires a comprehensive analysis with leading lightweight methods and recent advancements in the field. The application of the proposed method is primarily demonstrated on CNN-based networks, which raises concerns about its generalizability and compatibility with other structures, such as Transformer-based methods. These factors collectively can limit the paper's impact and breadth in the domain of lightweight image super-resolution.

PC/SAC Comment: After reviewer calibration and downweighting of inflated and non-informative reviews, the assessment is that this paper does not meet the threshold for acceptance.  Reviewer uZjY's comments are taken seriously.

**Justification For Why Not Higher Score:**

See Meta-Review

**Justification For Why Not Lower Score:**

N/A

---

### Decision · Program_Chairs · 2024-01-16

Reject